**PLOS** NEGLECTED TROPICAL DISEASES

# Accuracy of immunological tests on serum and urine for diagnosis of *Taenia solium* neurocysticercosis: A systematic review

Lisa Van Acker [1]*, Luz Toribio[2,3], Mkunde Chachage[4], Hang Zeng[5,6], Brecht Devleesschauwer[1,7], Héctor H. Garcia[3,8], Sarah Gabriël[1], on behalf of the NeuroSolve Consortium[¶]

1 Laboratory of Foodborne Parasitic Zoonoses, Department of Translational Physiology, Infectiology and Public Health, Faculty of Veterinary Medicine, Ghent University, Merelbeke, Belgium, 2 Infection and Immunity Institute, St George's University of London, London, United Kingdom, 3 Department of Microbiology, Universidad Peruana Cayetano Heredia, Lima, Peru, 4 Department of Microbiology and Immunology, University of Dar es Salaam-Mbeya College of Health and Allied Sciences, Mbeya, Tanzania, 5 School of Food and Bioengineering, Xihua University, Chengdu, China, 6 Key Laboratory of Food Microbiology of Sichuan, Xihua University, Chengdu, China, 7 Department of Health Information, Sciensano, Brussels, Belgium, 8 Bloomberg School of Public Health, Johns Hopkins University, Maryland, United States of America

¶ Membership of the NeuroSolve Consortium is provided in the Acknowledgements.
* lisa.vanacker@ugent.be

**Data Availability Statement:** All relevant data are within the manuscript and its Supporting Information files.

## Abstract

### Background

*Taenia solium* neurocysticercosis is a zoonotic neglected tropical disease, for which adequate diagnostic management is paramount, especially in patients with active cysts for whom improved and timely management could prove beneficial. Immunodiagnosis can potentially partially mitigate the necessity for neuroimaging, shortening the diagnostic -and treatment- pathway. An up-to-date review of immunological test performance is however lacking.

### Methodology/Principal findings

Searches were performed in PubMed, EMBASE, Web of Science, and Scopus (up to January 2024), with included records fitting the review scope, i.e. accuracy evaluation of an antibody-/or antigen-detecting immunological test, using serum or urine of humans confirmed via reference standard (i.e. neuroimaging or surgery/biopsy). Record data was assessed, with classification of descriptive data on cyst localization and stage according to a developed confidence scale, and with selection of tests evaluated on a sufficiently high sample size. A QUADAS-2 risk of bias assessment was performed. After screening, 169 records were included for data collection, with 53 records—corresponding to 123 tests- selected for analysis. Absence of data and large data heterogeneity complicated result interpretation. The lentil lectin-bound glycoprotein enzyme-linked immunoelectrotranfser blot seems to fulfill high accuracy standards regarding detection of parenchymal active multiple cysts; also antigen-detecting tests on serum and urine performed well, additionally in detection of

**Funding:** This study was carried out with support from NeuroSolve (https://neurosolve.net/), a European & Developing Countries Clinical Trials Partnership (EDCTP) project, Horizon Europe funded (HORIZON-JU-GH-EDCTP3). Grant Agreement ID: 101103306, project name: "Implementation of superior treatment regimen and improved patient pathway for neurocysticercosis in Sub-Saharan Africa". The funder had no role in study design, data collection and analysis, decision to publish, or preparation of the manuscript.

**Competing interests:** The authors have declared that no competing interests exist.

extraparenchymal neurocysticercosis. A novel multi-antigen print immunoassay is highly promising, with sensitivity for detection of extraparenchymal and parenchymal active single and multiple cysts of 100.0%, and specificity of 98.5%. Point-of-care tests showed promising results, however require further evaluation in targeted resource-poor settings.

## Conclusions/Significance

The review highlights the importance of transparent and unambiguous data reporting. With promising immunological tests in development, the challenge before usage in targeted settings will be to perform large-scale evaluations whilst holding into account both optimized test performance and ease of use. Accessibility to validated tests and feasibility of implementation should also be considered.

### Author summary

Neurocysticercosis is an important, but neglected disease in many low- and middle-income countries. In resource-poor areas, management of the disease is impeded by lack of availability of, and access to, adequate diagnostic techniques. Immunological tests, performed on serum or urine of affected humans, could be tools of interest in improving the diagnostic pathway. This systematic review provides an overview of immunological tests that have been evaluated so far, and especially focusses on test performance in detection of cysts with specific localization, stage and number. Results on test accuracy proved difficult to retrieve from published records. The comparison of obtained test results was exceedingly challenging due to large heterogeneity. With usable data meticulously selected, several known test formats, such as the LLGP-EITB and the antigen ELISA, showed expected performance results, and some novel test formats, such as the multi-antigen print immunoassay, were highly promising. Also, urine-based tests could provide a non-invasive alternative to serum-based tests. Evaluation of immunological tests in non-clinical settings requires a sufficient sample size for further analysis of data. To improve management of the disease in targeted resource-poor settings, immunological test formats will have to comply with high performance and ease-of-use standards, to optimize chances of future implementation.

## Introduction

*Taenia solium*, or the pork tapeworm, is a zoonotic foodborne parasite. The neglected tropical disease taeniasis/cysticercosis caused by this parasite, is endemic in various areas including areas in Africa, Latin America, and South- and South-East Asia. Its presence is however not restricted to these regions, as imported cases are sporadically seen in high-income non-endemic countries [1–3]. The main public health concern and leading cause of acquired epilepsy worldwide, lies in the infection of humans with the metacestode larval stage, which develops as a cyst in the central nervous system—referred to as neurocysticercosis (NCC) [4]. Cysts can be localized in the brain parenchyma, or, less frequently, have an extraparenchymal localization, mainly in the subarachnoid or ventricular space. Different cyst stages are discernible, indicating the cyst's progression from viable (i.e. active), over degenerating, to a calcified stage

(i.e. inactive) [5]. Clinical manifestations of NCC range from severe progressive headache and epileptic seizures, to hydrocephalus, vasculitis and arachnoiditis [6,7].

Paramount in NCC management is the timely and adequate diagnosis of the disease, most importantly diagnosis of cysts in the active stage, ensuring an equally early initiation of treatment. Various factors complicate the diagnosis of NCC. Symptoms often only arise months to years after infection, and clinical manifestations are highly unspecific and depend on many factors such as cyst localization, number, size and stage; therefore emphasizing the necessity of neuroimaging confirmation [7,8]. Furthermore, endemic regions for NCC harbour many other infectious diseases with similar symptoms and/or antigenic components to *T. solium* cysticercosis, thus increasing the chance of test cross-reactions [9]. To aid in diagnosis of the disease, two sets of criteria have been developed, i.e. the criteria by Del Brutto *et al.* [10,11] and the criteria by Carpio *et al.* [12]. In both, neuroimaging modalities (computed tomography (CT) and magnetic resonance imaging (MRI)) and serological tests for neuroimaging selection and confirmation (antibody-detecting enzyme-linked immunoelectrotranfser blot (EITB), or antigen-detecting enzyme-linked immunosorbent assay (ELISA)), are deemed indispensable to achieve accurate diagnosis. However, in endemic and resource-poor settings, neuroimaging modalities are often dysfunctional, inaccessible or unavailable. Also, the use of these modalities is costly, and requires highly-trained staff capable of interpreting images of varying quality depending on the obtainable image resolution [13]. Similar reservations can be made for immunological tests, although to a far lesser extent. Currently, hindrances such as availability, adaptability and low or unknown accuracy still impede the uptake and use of these tests. In easy-to-use and cost-saving test formats, immunological tests detecting specific antibodies or antigens, could form the key to reduce the need of neuroimaging, hereby improving NCC diagnostic -and treatment- management. To date, numerous immunological tests for detection of NCC have been reported, however, first a clear picture is needed of the currently available test formats and their performance. Regarding serological tests for antibody detection, such as Western blot and antibody ELISA, the LLGP (lentil lectin-bound glycoprotein)-EITB is known as the assay of preference in clinical settings due to its high test accuracy [14]. However, it requires experienced lab personnel, intricate equipment and a large amount of parasite material to produce test strips. Many commercial tests are based on the ELISA format using crude or semi-purified antigens, but these do not score well on diagnostic performance [15]. Immunodiagnosis has evolved in recent years, with a shift from use of crude or semi-purified antigens, to the use of recombinant and synthetic antigens for detection of antibodies against NCC. Serological antigen-detecting tests, such as the antigen ELISA, have demonstrated to be excellent tools for detection of active NCC infections [16]. Interestingly, the use of urine as a parasite antigen source has also been explored in the development of immunological tests, providing an alternative to tests that depend on more invasive blood collection [17,18]. Tests using cerebrospinal fluid samples were not included in this review, as they require an even more invasive approach and are therefore not applicable in field conditions. Also point-of-care (POC) tests provide significant advantages in terms of usability and cost-effectiveness, but still lack adequate accuracy results when used in hospital-/community-based settings [19,20].

In 2017, the WHO published several updated Target Product Profiles (TPPs) for diagnosis of *T. solium*, providing key characteristics of diagnostic tools for the development of products applicable in specific settings [21]. The call for applicable, easy-to-use, and cost-effective diagnostic tools is evident, with a special need for tools that are adaptable to resource-poor settings. A previously published systematic review on immunodiagnosis of NCC, by Cardona-Arias *et al.* 2017 [22], provides an overview of existing tests, however, since then, new developments and evaluations have occurred. Our systematic review carefully extends the overview of characteristics and performance of existing immunological tests and -in extent- of antibodies or

antigens utilized in these tests (hereon forward referred to as the *diagnostic reagent*), focusing on biological serum and urine samples. Additionally, test performance was evaluated according to the localization, stage and number of cysts.

## Methods

### Review questions and search syntax

A systematic review was conducted of published literature, and reported according to the Preferred Reporting Items for Systematic Review and Meta-Analysis (PRISMA) guidelines [23]. A PRISMA checklist can be found in the Supplementary Materials (S1 Checklist). The following review questions were posed: How do different immunological tests, and utilized diagnostic reagents, on biological serum/urine samples for diagnosis of neurocysticercosis perform regarding accuracy? What is the performed accuracy of immunological tests for diagnosis of neurocysticercosis with different cyst localization, stage, and number? In order to identify relevant records, a search syntax of Boolean operators (AND, OR, NOT, *) and key words involving "(human) (neuro)cysticercosis", "immunological tests", "serological tests", "diagnostic marker", "accuracy", and "serum"/" urine" was composed. The search syntax was applied in four different search engines (i.e. PubMed, EMBASE, Web of Science and Scopus) without restrictions of language or publication date (S1 Search Strategy). Obtained records from databases were merged in reference management software EndNote 20 [24].

### Record selection

Three screening phases were performed to acquire relevant records for this systematic review. The first screening phase was performed in EndNote, for removal of duplicate records. The second screening phase was performed using the web tool Rayyan [25], for selection of eligible records according to pre-defined eligibility criteria, by title and abstract (TIAB) screening. This screening was performed independently and blinded by two authors in case of English records (LVA, LT), and by one author in case of Spanish and Portuguese (LT), or Chinese (HZ) records. The third screening phase was again conducted in Rayyan, now with full-text eligibility screening, by one author (respectively LVA for English, LT for Spanish and Portuguese, and HZ for Chinese records). Screenings were performed according to following exclusion criteria: (i) records with no available full text, (ii) records with an incorrect publication and/or study type (defined in S1 Protocol), (iii) records not concerning an immunological test or diagnostic reagent for detection of antigens or antibodies in NCC diagnosis, (iv) records with immunodiagnosis not performed on samples originating from humans with NCC confirmed via reference standard(s), i.e. neuroimaging and/or surgery/biopsy, (v) records with immunodiagnosis not performed on urine and/or serum samples of infected patients (including whole blood and plasma), and (vi) records not -as a main focus- evaluating accuracy of an immunological test/diagnostic reagent. As test accuracy evaluation by localization, stage and number of cysts was regarded of high importance in this review, records reporting data on test specificity only, and not on test sensitivity, were not included. When unclarities were encountered regarding inclusion or exclusion of records for the third screening phase, corresponding authors were contacted. Records not identified via the search syntax, were sought out via backward snowballing (i.e. accessing the reference lists of each selected record, and assessing eligibility of the reference list records by reading the full text). One additional submitted, preprinted record was added before submission of this paper, obtained via internal communication. Final selected records all fitted the scope of the review, evaluating accuracy of an antibody-/or antigen-detecting immunological index test, with use of serum, urine, plasma or

whole blood samples, of humans confirmed to have NCC via reference standard (i.e. neuroimaging and/or surgery/biopsy).

Data collection was performed in a Microsoft Excel macro-enabled worksheet by one author (respectively LVA for English, LT for Spanish and Portuguese, and HZ for Chinese records), and with a quality control of 10% of English records by an additional author (MC). Per record, data was collected on (i) study characteristics, (ii) study participants, (iii) test samples, (iv) immunological test, and (v) additional information. Study participants were subdivided into three categories: 1) NCC group (consisting of patients with confirmed NCC via defined reference standard(s)), 2) control group (consisting of healthy individuals, or individuals with other non-infectious neurological conditions), and 3) other infections group (consisting of individuals with infections other than *T. solium* cysticercosis). Data on immunological tests included test characteristics, test performance, cross-reactivity, test development stage, and availability. Further information on the review methodology is detailed in the published protocol of the systematic review on the International Prospective Register of Systematic Reviews (PROSPERO) (https://www.crd.york.ac.uk/prospero/display_record.php?ID=CRD420 23440930) (S1 Protocol).

Study quality was assessed via a Risk of Bias (RoB) assessment, applying the Quality Assessment of Diagnostic Accuracy Studies-2 (QUADAS-2) tool [26], adapted to the current review (S1 Protocol). The RoB assessment was done by one author (respectively LVA for English, LT for Spanish and Portuguese, and HZ for Chinese records). Sixteen signalling questions were applied for risk assessment over four domains, i.e. Patient selection (Q1-4), Index test(s) (Q5-9), Reference standard(s) (Q10-13), and Flow and Timing (Q14-16) (S1 Text). Signalling questions were formulated in a way that a positive answer is indicative of a low risk of bias, whereas a negative answer is indicative of a high risk of bias.

## Data classification and reporting

A confidence scale was developed to assess the usability of collected data, in terms of availability of descriptive data on cyst localization and cyst stage (Fig 1). For cyst localization, data was classified into one of three groups, i.e. parenchymal (only parenchymal cysts), extraparenchymal (only extraparenchymal cysts localized in either subarachnoid or ventricular space, or not specified), or parenchymal + extraparenchymal (a combination of both). For cyst stage, descriptive data was matched to one of two groups, i.e. active (viable cysts and/or cysts in transitional stage, with or without additional inactive cysts), or inactive (only calcified cysts). Based on developed criteria as can be consulted in Fig 1, a level of certainty (either 'definite', or 'probable', or 'possible') was assigned to each evaluated entity, regarding both cyst localization and stage. With both certainty levels regarded 'definite', or one 'definite' and one 'probable', the data of the evaluated test was included for analysis with high confidence. Those with both levels 'probable', or one 'definite' with one 'possible', were included with low confidence. When data on test accuracy was inadequate (e.g. data on cyst localization and cyst stage was not provided, or data was provided but insufficient for assignment of certainty levels, or certainty levels were assigned but data was unmatchable), or when data did not fulfill the criteria for abovementioned classification, the concerned entity was excluded from further analysis.

Included tests with high confidence were individually placed into one of following categories: serological antibody detection, serological antigen detection, urine-based antibody detection, urine-based antigen detection. Within each category, a further classification was made according to the test format of used index test (e.g. Western blot) and according to the diagnostic reagent (e.g. *T. solium* somatic antigen: cyst fluid), grouping entities based on the same type of index test and a similar origin of diagnostic reagent. If not clearly specified on which

| **Cyst localization: PARENCHYMAL** | |
| --- | --- |
| Certainty level | Criteria |
| Definite | specific mention of *parenchymal* localization (cerebral cortex - cortical lobes, basal ganglia), *benign,* any reference to specific extraparenchymal subarachnoid localization of cysts in the cortical sulci/ convexity (parenchymal-like) |
| Probable | single cyst granuloma (SCG), any reference to inactive cyst phase |
| Possible | small cyst(s) (<2mm diameter), symptoms: seizures/ headache |

| **Cyst localization: EXTRAPARENCHYMAL** | | | |
| --- | --- | --- | --- |
| Certainty level | Criteria | | |
| | **SUBARACHNOID** | **INTRAVENTRICULAR** | **UNKNOWN / COMBINATION** |
| Definite | specific mention of *extraparenchymal subarachnoid* or *subarachnoid* presence of cysts (basal cisterns, Sylvian fissures), or *racemose* (agglomerating) | specific mention of *extraparenchymal (intra)ventricular* or *(intra)ventricular* presence of cysts | specific mention of *extraparenchymal* localization of cysts, *malignant* |
| Probable | presence of *arachnoiditis, leptomeningeal enhancement, meningitis, meningoencephalitis, ependymitis, vasculitis* | signs of ventricular enlargement | symptoms: increased intracranial pressure (IICP)/ hydrocephalus |
| Possible | / | / | large cyst(s) (>2mm diameter) |

| **Cyst localization: PARENCHYMAL + EXTRAPARENCHYMAL** | |
| --- | --- |
| Certainty level | Criteria |
| Definite | At least one criterion of PARENCHYMAL and one criterion of EXTRAPARENCHYMAL, in the combination 'Definite' + 'Definite' |
| Probable | At least one criterion of PARENCHYMAL and one criterion of EXTRAPARENCHYMAL, in the combination 'Probable' + 'Definite'/'Probable' |
| Possible | At least one criterion of PARENCHYMAL and one criterion of EXTRAPARENCHYMAL, in the combination 'Possible' + 'Definite'/'Probable'/'Possible' |

| **Cyst stage: ACTIVE** | |
| --- | --- |
| Certainty level | Criteria |
| Definite | *active, viable, live, intact, vesicular, acute, rounded/regular/well-delineated lesions, low-density lesions, scolex visible, degenerating, transition(al), colloidal vesicular, colloidal, granular-nodular, granular, (ring-)enhancing lesions, (nodular) enhancing lesions, single cyst granuloma (SCG),* multiple stages/phases (with or without additional criteria for INACTIVE) |
| Probable | extensive inflammatory response/ oedema, any reference to extraparenchymal localization (with or without additional criteria for INACTIVE) |
| Possible | symptoms: seizures/ headache (with or without additional criteria for INACTIVE) |

| **Cyst stage: INACTIVE** | |
| --- | --- |
| Certainty level | Criteria |
| Definite | *inactive, calcified, calcification, mineralized, mineralization, degenerated, nodular, nodules* |
| Probable | / |
| Possible | / |

| | | Certainty level for cyst localization (PARENCHYMAL / EXTRAPARENCHYMAL / PARENCHYMAL+EXTRAPARENCHYMAL) | | | |
| --- | --- | --- | --- | --- | --- |
| | | Definite | Probable | Possible | No indication |
| Certainty level for cyst stage (ACTIVE / INACTIVE) | Definite | Include with high confidence | Include with high confidence | Include with low confidence | Exclude |
| | Probable | Include with high confidence | Include with low confidence | Exclude | Exclude |
| | Possible | Include with low confidence | Exclude | Exclude | Exclude |
| | No indication | Exclude | Exclude | Exclude | Exclude |
| | | Exclude if needed data is insufficient | | | |

**Fig 1. Proposed certainty criteria and classification grid for confidence scaling of evaluated entities.**

diagnostic reagent the index test was based (e.g. no clarification of the type of somatic antigen), the corresponding author of the record was contacted for enquiry. If no clarification was provided, the entity was excluded (due to insufficient data for categorization). If multiple diagnostic reagents were tested within a record (e.g. multiple recombinant and synthetic antigens), followed by combining the best performing fractions, only the test evaluating the best performing combination was included. For further classification, the cyst number was considered in case of parenchymal localization (i.e. parenchymal single cysts and/or parenchymal multiple cysts). When not mentioned, the test was excluded, unless data was provided on extraparenchymal localization of cysts, regardless of cyst number. Next, the added sample size of entities within a subgroup with the same type of index test and a similar origin of diagnostic reagent was considered (e.g. the subgroup of Western blots based on *T. solium* somatic antigen: cyst fluid). Added sample sizes were calculated per following class of cyst types: parenchymal active, parenchymal inactive, extraparenchymal, parenchymal + extraparenchymal. If, in a subgroup, the added sample size in a cyst type class was below 20, the concerning data of this class was excluded from analysis to reduce the risk of bias and ensure a more reliable and consistent interpretation of the evidence. A similar procedure was followed to categorize and select low confidence data.

For each included test, data on test sensitivity according to cyst localization, stage and number was reported, depending on data availability. If available, data on test specificity was also added, i.e. specificity for the control group, for the other infections group, and/or for a combination of both. As substantial data heterogeneity was observed, narrative analysis was preferred to meta-analysis.

## Results

### Search results

A total of 2315 records were identified through database searches (performed on 4th January 2024). This included 569, 692, 557, and 497 records from PubMed, EMBASE, Web of Science, and Scopus, respectively. During the first screening stage, 1400 duplicate records were removed. In further screening phases, 700 records (second screening) and 63 records (third screening) were excluded, as they did not meet our eligibility criteria, while 16 records were introduced via scanning of reference lists. An additional preprinted paper was added [27]. This consecutively led to 169 records that were included for data collection, of which full-text could be found in following languages: English (159 records), Spanish (6 records), Chinese (2 records), and Portuguese (2 records). Of 159 English records, 10% (corresponding to 16 records) were randomly selected for quality control to evaluate data extraction reliability. No systematic errors were perceived in the data collection process. An overview of the record selection process, including defined eligibility criteria, was constructed as a PRISMA flowchart diagram (Fig 2). Before finalization of the review, database searches were repeated on 30th April 2024, with no additional inclusion of records according to eligibility criteria.

### Risk of bias assessment

Results of the RoB analysis performed on selected studies, using QUADAS-2 modified signalling questions, are presented in Fig 3. Within the domain of Patient selection, many negative responses were recorded to Q2 (i.e. avoidance of a case-control design), indicating a high risk of selection bias when enrolling patients in a case group or control group before performing the index test(s). The allocation of patients to a case group based on affirmative reference standard results, was predefined as an eligibility criterium for study selection. The allocation of patients to a control group, however, when performed without confirmation via reference

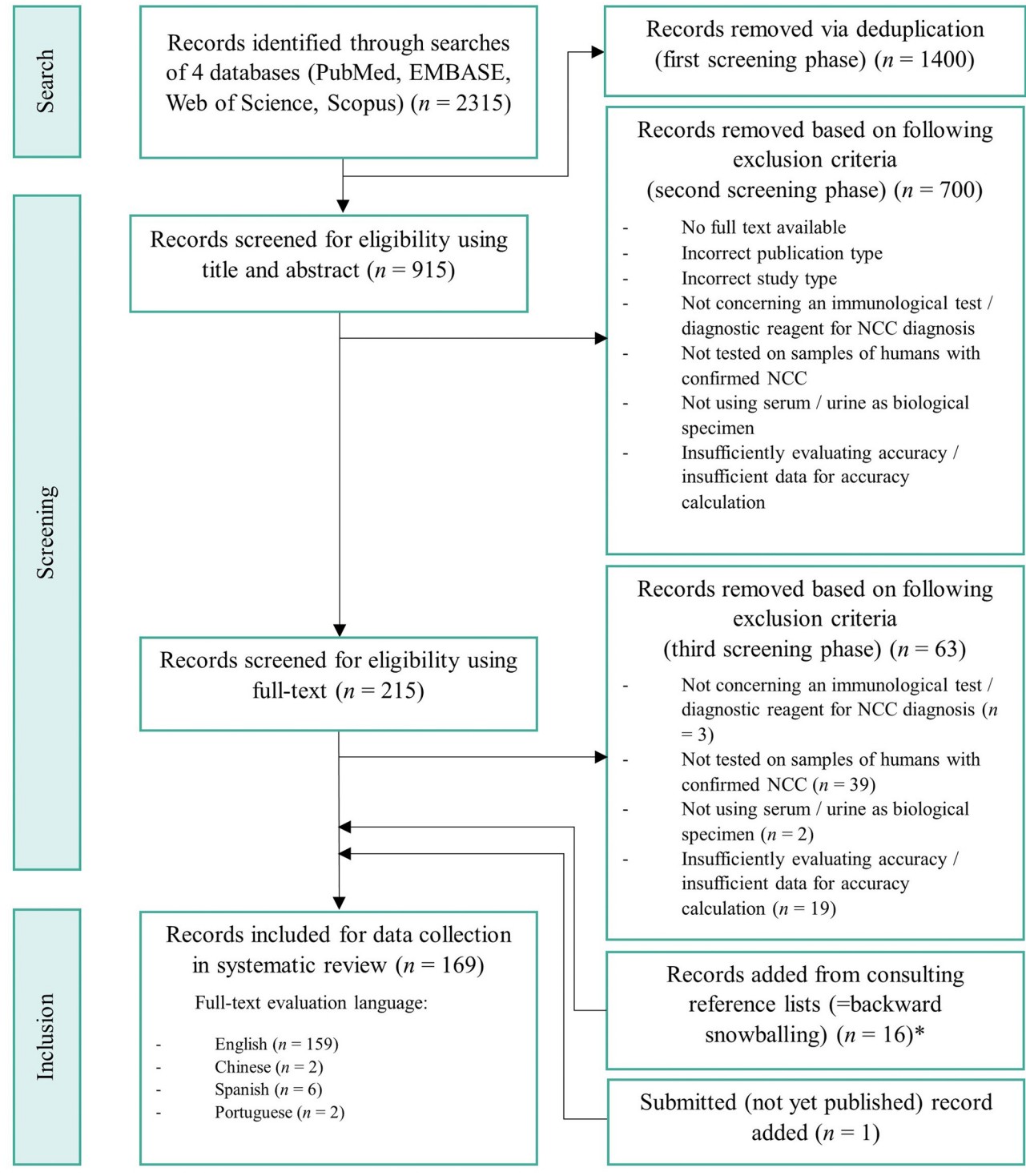

* One of these records had been removed from selection during the second screening phase of database identified records. However, it was reincluded based on title during backward snowballing, and reinstalled in selection based on full-text screening.

**Fig 2. Preferred Reporting Items for Systematic Review and Meta-Analysis (PRISMA) flowchart diagram of the record selection process.**

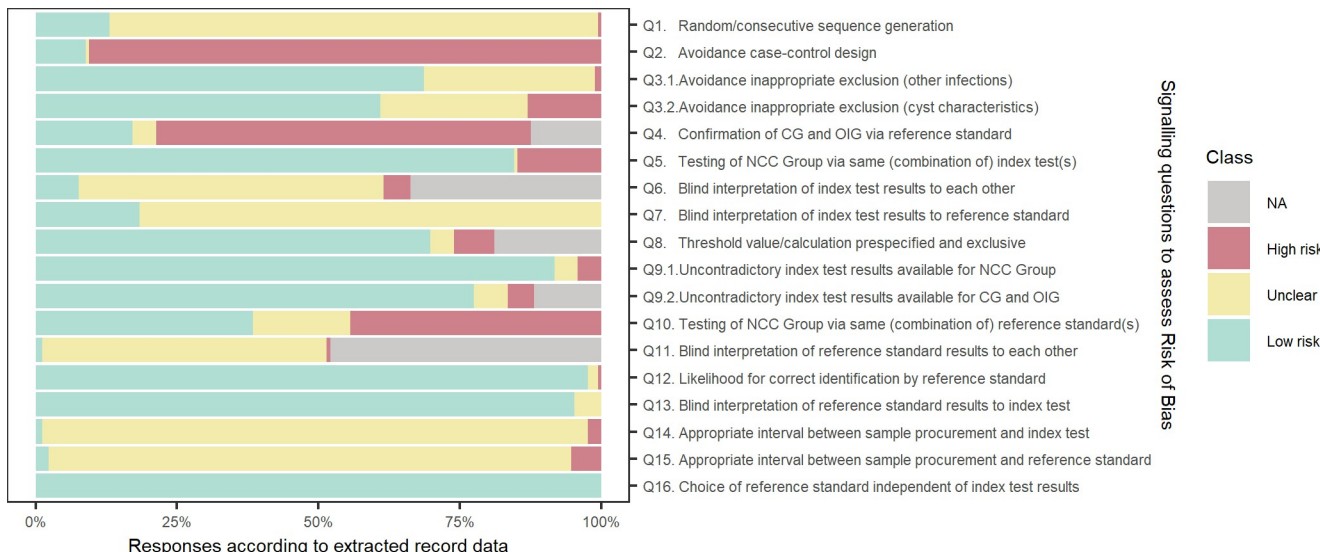

**Fig 3. Risk of Bias assessment using QUADAS-2 tool modified signalling questions.**

standard (Q4), contributed to potential risk of selection bias. With the reference standard often performed preliminary to the index test(s), blinding to index test results is highly likely, as was asserted by many positive responses to Q13 (i.e. blind interpretation of reference standard results to the index test). On the other hand, blind interpretation of index test results to reference standard results must be ensured. Missing record data on blinding, however, led to increase in index test risk of bias (Q7). Lacking data also strongly affected selection bias (Q1), and flow and timing risk of bias (Q14-15). A summary of the results of the RoB assessment for all included studies can be found in Supplementary Materials (S1 Table).

## Data analysis

Data classification was performed to select studies providing adequate data on cyst localization and cyst stage. Of 169 records included for data collection, corresponding to 504 evaluated tests -as some records evaluate more than one index test-, 116 records corresponding to 381 tests were excluded according to exclusion criteria (Fig 4).

Records and related tests included with high and low confidence, are displayed in Tables 1 and 2, respectively. In both tables, records were summarized by test format, used biological specimen, and diagnostic reagent (as specified in the record, or clarified via communication with the author). A total of 53 studies were included in both tables, with 123 tests evaluated. Of these 53 studies, 40% (21/53) were conducted in South America [17,28–47], 23% (12/53) in South Asia [48–59], 11% (6/53) in East Asia [60–65], 9% (5/53) in North America including Mexico [66–70], 6% (3/53) in Eastern Africa [20,27,71], 2% (1/53) in Western Africa [72], 2% (1/53) in Central America [73], and 8% (4/53) not specified [74–77]. Of 123 included tests, evaluated test accuracy results concerning both sensitivity and specificity were available for 106 tests, whereas 17 of included tests reported sensitivity alone. Test results are presented as a percentage, with additional data displayed as a proportion (i.e. regarding sensitivity, the proportion of true positives / (true positives + false negatives), and regarding specificity, the proportion of true negatives / (true negatives + false positives)).

Despite considerable data exclusion to reserve data confidence, several patterns can be identified within the included test results demonstrated in Table 1. The Western Blot based on

| | # excluded records | # excluded tests |
|---|---|---|
| No descriptive data and/or no corresponding accuracy data provided for cyst localization/stage | 57 | 178 |
| Unmatchable accuracy data when combining available descriptive data | 1 | 3 |
| Available descriptive data is insufficient to assign a certainty level for cyst localization/stage | 4 | 8 |
| A certainty level is assigned for either cyst localization or cyst stage, not for both | 20 | 63 |
| The combination of both certainty levels (for cyst localization and cyst stage) leads to exclusion | 0 | 0 |
| Diagnostic reagent unspecified | 1 | 2 |
| A different test is used to evaluate the combination of best performing fractions (diagnostic reagents) | 0 | 10 |
| Cyst number not defined in case of parenchymal localization (no data on extraparenchymal localization) | 30 | 108 |
| Added sample sizes for defined cyst type class within a subgroup <20 | 3 | 9 |
| | **116** | **381** |

**Fig 4. Exclusion criteria for data analysis, with amount of excluded records and corresponding tests.**

LLGP shows sensitivities of 81.1–100.0% in detection of parenchymal active multiple cysts [31,32,53], and in all but one study [31], sensitivity drops significantly for single cysts (<62.6%) [51–54]. The expected high specificity for the Western Blot was also confirmed (92.3–100.0%) [20,29,31,51,52,54,66,67,73]. Although a significant number of studies using the antibody ELISA test format have been included, data is largely variable, and insufficient to make comparisons with other test formats. Regarding the use of recombinant and/or synthetic antigens as test diagnostic reagents, results were variable. The newly developed multi-antigen print immunoassay (MAPIA) based on three recombinant and synthetic antigens, shows notable sensitivity for parenchymal active single and multiple cysts (100.0%) and extraparenchymal cysts in the subarachnoid space (100.0%), and delivers high specificity (98.5%) [31]. Although these results are based on small sample sizes, the MAPIA has potential to form an interesting addition to standard laboratory performed tests. The Western Blot based on rTsEndoB1 [60], and the Western Blot based on rTsAg5 [33], display promising sensitivities for detection of parenchymal active (71.9%) [33] and inactive (72.5%) [60] multiple cysts and extraparenchymal cysts (96.4%) [33], however limited specificity in the control group (75.7%) [33], and cross-reactivities (e.g. with taeniasis, hydatidosis) could limit their use [60]. Regarding serological antigen-detecting tests, based on monoclonal antibodies such as B158/B60 and TsW8/TsW5, high sensitivity values were observed in case of parenchymal active multiple cysts (75.0%, 100.0%) [27,71], and extraparenchymal subarachnoid cysts (97.8–100.0%) [30,43]. Similarly, urine-based antigen-detecting tests (antigen ELISA and POC dipstick) presented sensitivities of 96.2% and 96.7% for parenchymal active multiple cysts and extraparenchymal subarachnoid cysts, respectively [17,46].

Some reports that evaluated commercial kits, i.e. the QualiCode Cysticercosis Western Blot Kit developed by Immunetics Inc. [66,72], the Cysticercosis Western Blot IgG developed by LDBIO Diagnostics [67], the NovaLisa Taenia solium IgG developed by NovaTec Immundiagnostica GmbH [67], and the Cysticercosis Ag ELISA developed by apDia [43], presented too little data to confirm whether or not commercial kits deliver inadequate test performance.

Table 1. Test sensitivity and specificity of serological and urine-based antibody- and antigen-detecting tests, included with high confidence, categorized per test format and diagnostic reagent.

| Test format | Diagnostic reagent | RS# | Record | par act 1* | par act mult* | par inact 1* | par inact mult* | expar sub* | expar ven* | expar (NS)* | par +expar act* | CG** | OIG** | CG + OIG** | Cross-reactivity |
|---|---|---|---|---|---|---|---|---|---|---|---|---|---|---|---|
| **Serological antibody detection** | | | | | | | | | | | | | | | |
| **Western Blot** | *T. solium* somatic Ag: WCE | | | | | | | | | | | | | | |
| | whole cyst Ag | MRI | Arthi 2021 [48] | 44.4 (8/18) | - | - | - | - | - | - | - | - | - | - | - |
| | low-molecular mass 10-30kDa | CT, MRI | Atluri 2009a [49] | 79.8 (87/109) | - | - | - | - | - | - | - | 75.3 (64/85) | 45.0 (18/40) | 65.6 (82/125) | ml NS/10, tp NS/7, hd NS/9, am NS/12, as NS/2 |
| | low-molecular mass 10-30kDa | CT, MRI | Atluri 2011 [50] | 100.0 (11/11) | - | - | - | - | - | - | - | 100.0 (17/17) | 75.0 (6/8) | 92.0 (23/25) | hd 2/3, tp 0/2, ml 0/1, am 0/1, as 0/1 |
| | total saline extract | imaging | Barcelos 2007 [28] | 100.0 (2/2) | 100.0 (8/8) | - | - | - | - | - | - | 100.0 (10/10) | - | - | - |
| | *T. solium* somatic Ag: CF | | | | | | | | | | | | | | |
| | crude cyst fluid | CT, MRI | Bae 2008 [74] | - | - | - | 42.5 (17/40) | - | - | - | - | 100.0 (60/60) | 78.7 (166/211) | 83.4 (226/271) | Tsot 3/14, Tsag 3/20, Tas 3/25, ae 6/15, ce 19/37, sp 2/20, pg 2/20, cl 4/20, fs 1/20, ss 2/20 |
| | chimera 120kDa + 150 kDa (CF) | CT, MRI | Bae 2008 [74] | - | - | - | 32.5 (13/40) | - | - | - | - | 100.0 (60/60) | 97.2 (205/211) | 97.8 (265/271) | Tsot 1/14, Tsag 0/20, Tas 0/25, ae 0/15, ce 3/37, sp 1/20, pg 0/20, cl 0/20, fs 1/20, ss 0/20 |
| | IEF-purified cyst Ag (CF) | CT, MRI | Oommen 2004 [51] | 17.1 (7/41) | - | - | - | - | - | - | - | 92.3 (24/26) | - | - | - |
| | *T. solium* ESP | | | | | | | | | | | | | | |
| | excretory secretory Ag | CT, MRI | Atluri 2009a [49] | 85.3 (93/109) | - | - | - | - | - | - | - | 76.5 (65/85) | 37.5 (15/40) | 64.0 (80/125) | ml NS/10, tp NS/7, hd NS/9, am NS/12, as NS/2 |
| | *T. solium* LLGP | | | | | | | | | | | | | | |
| | test kit Ag (Immunetics Inc, Cambridge, MA) | CT, MRI | Aguilar-Rebolledo 2002 [66] | - | - | - | - | 0.0 (0/5) | - | - | - | 96.0 (48/50) | - | - | - |
| | LLGP cyst Ag (InDRE) | NS | Hernández 2019 [67] | - | - | - | - | - | - | 89.7 (26/29) | - | 95.1 (39/41) | - | - | - |
| | LLGP cyst Ag (CDC) | NS | Hernández 2019 [67] | - | - | - | - | - | - | 93.1 (27/29) | - | - | - | - | - |
| | test kit Ag (LDBIO Diagnostics, Lyon, France) | NS | Hernández 2019 [67] | - | - | - | - | - | - | 100.0 (21/21) | - | - | - | - | - |

(*Continued*)

**Table 1.** (Continued)

| Test format | Diagnostic reagent | RS# | Record | par act 1* | par act mult* | par inact 1* | par inact mult* | expar sub* | expar ven* | expar (NS)* | par +expar act* | CG** | OIG** | CG + OIG** | Cross-reactivity |
|---|---|---|---|---|---|---|---|---|---|---|---|---|---|---|---|
| | LLGP cyst Ag (CDC) | CT, MRI | Oommen 2004 [51] | 60.0 (27/45) | - | - | - | - | - | - | - | 92.3 (12/13) | - | - | - |
| | LLGP cyst Ag (CDC) | CT | Palacio 1998 [29] | - | - | - | 36.4 (12/33) | - | - | - | - | 98.0 (350/357) | - | - | - |
| | LLGP cyst Ag | CT, MRI | Prabhakaran 2004 [52] | 62.6 (67/107) | - | - | - | - | - | - | - | 97.0 (97/100) | 100.0 (7/7) | 97.2 (104/107) | fl 0/2, tp 0/3, ra 0/2 |
| | LLGP cyst Ag | CT, MRI | Rodriguez 2009 [30] | - | - | - | - | 100.0 (31/31) | 100.0 (12/12) | - | - | - | - | - | - |
| | LLGP cyst Ag | CT | Sánchez 1999 [73] | - | - | 20.0 (3/15) | 38.9 (7/18) | - | - | - | - | 92.9 (13/14) | - | - | - |
| | LLGP cyst Ag | CT | Schantz 1994 [68] | 57.9 (176/304) | 81.1 (90/111) | 71.4 (5/7) | 100.0 (6/6) | - | - | - | - | - | - | - | - |
| | LLGP cyst Ag | CT | Stelzle 2024 [20] | | | 0.0 (0/3) | 25.0 (5/20)†1 | - | - | - | - | 97.0 (NS) | - | - | - |
| | LLGP cyst Ag | CT, MRI | Toribio 2023a [31] | 100.0 (29/29) | 100.0 (9/9) | - | - | 100.0 (40/40) | - | - | - | 100.0 (68/68) | - | - | - |
| | LLGP cyst Ag | CT, MRI | Vasudevan 2022 [53] | 57.9 (176/304) | 81.1 (90/111) | 34.8 (23/66) | 57.5 (23/40) | - | - | - | - | - | - | - | - |
| | LLGP cyst Ag | CT, MRI | Zea-Vera 2013 [32] | - | 100.0 (6/6) | - | - | - | 100.0 (1/1) | - | - | - | - | - | - |
| | non-solubilized LLGP | CT, MRI | Oommen 2004 [51] | 62.6 (67/107) | - | - | - | - | - | - | - | 97.0 (97/100) | - | - | - |
| | urea-induced conformed LLGP | CT, MRI | Prabhakaran 2007 [54] | 46.7 (28/60) | - | - | - | - | - | - | - | 96.0 (38/40) | - | - | - |
| | *T. solium* recombinant Ag | imaging | Ahn 2016 [60] | - | - | - | 72.5 (74/102) | - | - | - | - | 100.0 (75/75) | 58.2 (246/423) | 64.5 (321/498) | ce 81/101, ae 48/61, Tsot 7/11, Tsag 8/12, Tas 8/12, dp 3/14, sp 22/92, Ssj 0/20, pg 0/20, cl 0/20, an 0/20, tr 0/20, tp 0/10, am 0/10 |
| | r chimera (18kDa (120kDa CF) + b1- + RS-1 + m13h-variant (150kDa CF) | CT, MRI | Bae 2008 [74] | - | - | - | 47.5 (19/40) | - | - | - | - | 100.0 (60/60) | 96.7 (204/211) | 97.4 (264/271) | Tsot 1/14, Tsag 0/20, Tas 0/25, ae 0/15, ce 3/37, sp 1/20, pg 1/20, cl 0/20, fs 1/20, ss 0/20 |

*(Continued)*

**Table 1.** (Continued)

| Test format | Diagnostic reagent | RS# | Record | par act 1* | par act mult* | par inact 1* | par inact mult* | expar sub* | expar ven* | expar (NS)* | par +expar act* | CG** | OIG** | CG + OIG** | Cross-reactivity |
|---|---|---|---|---|---|---|---|---|---|---|---|---|---|---|---|
| | rTsMFas1 | imaging | Bae 2014 [75] | - | - | - | 78.8 (63/80) | - | - | - | - | 100.0 (50/50) | 91.6 (228/249) | 93.0 (278/299) | Tsot 1/30, ae 2/33, ce 10/56, sp 4/50, Ssj 2/30, cl 2/50 |
| | r10kDa (150kDa CF) | CT, MRI | Chung 1999 [61] | - | - | - | 14.0 (4/29) | - | - | - | - | 100.0 (50/50) | 97.7 (127/130) | 98.3 (177/180) | ae 1/11, ce 0/9, sp 1/30, pg 0/30, cl 1/30, fs 0/10, Ssj 0/10 |
| | r18kDa (120kDa CF) | CT, MRI | Lee 2005a [62] | - | - | - | 20.0 (2/10) | - | - | - | - | 100.0 (NS) | NS (NS) | 97.1 (34/35) | sp 1/NS, ae 0/NS, ce 0/NS |
| | rTsAg5 | imaging | Rueda 2011 [33] | 38.7 (21/53) | 71.9 (41/57) | - | - | - | - | 96.4 (53/55) | - | 75.7 (78/103) | 91.4 (85/93) | 83.2 (163/196) | Hmn 6/28, Dpp 0/2, Tsag 1/10, ce 1/16, Asl 0/2, Env 0/8, Trt 0/6, anc 0/9, Sts 0/12 |
| | rT24H | CT | Stelzle 2024 [20] | - | - | 0.0 (0/3) | 50.0 (10/20)†2 | - | - | - | - | 98.0 (NS) | - | - | - |
| | rT24H | CT | Zulu 2024 [27] | - | - | 9.1 (1/11) | 42.1 (8/19)†3 | - | - | - | - | 89.0 (NS) | - | - | - |
| **ab ELISA** | *T. solium* somatic Ag: WCE | | | | | | | | | | | | | | |
| | crude soluble extract | CT, MRI | Atluri 2009b [55] | 38.5 (42/109) | - | 0.0 (0/2) | 0.0 (0/3) | - | - | - | - | NS (NS/85) | NS (NS/40) | 88.0 (110/125) | ml NS/10, tp NS/7, hd NS/9, am NS/12, as NS/2 |
| | low-molecular mass 10-30kDa | CT, MRI | Atluri 2009b [55] | 66.0 (72/109) | - | 50.0 (1/2) | 0.0 (0/3) | - | - | - | - | NS (NS/85) | NS (NS/40) | 85.6 (107/125) | ml NS/10, tp NS/7, hd NS/9, am NS/12, as NS/2 |
| | whole cysticerci | CT, s/b | Corona 1986 [76] | - | - | - | - | - | - | 93.0 (29/31) | - | 90.0 (107/119) | - | - | - |
| | whole cysticerci | CT, s/b | Mohammad 1984 [69] | - | - | - | - | - | 100.0 (5/5) | - | - | 100.0 (19/19) | - | - | - |
| | crude saline extract | CT | Schantz 1994 [68] | - | - | 42.9 (3/7) | 0.0 (0/6) | - | - | - | - | - | - | - | - |
| | *T. solium* somatic Ag: CF | | | | | | | | | | | | | | |
| | vesicular fluid | CT, MRI | Arruda 2005 [34] | - | - | - | - | - | - | - | 0.0 (0/1) | 100.0 (48/48) | 90.6 (29/32) | 96.2 (77/80) | sy 1/6, tp 0/6, Lsd 0/3, Ssm 0/4, Ssh 0/2, cm 1/2, im 0/3, hepA 0/2, hepB 1/4 |
| | crude cyst fluid | CT, MRI | Bae 2008 [74] | - | - | - | 47.5 (19/40) | - | - | - | - | 100.0 (60/60) | 76.3 (161/211) | 81.5 (221/271) | Tsot 4/14, Tsag 3/20, Tas 2/25, ae 5/15, ce 21/37, sp 3/20, pg 3/20, cl 4/20, fs 3/20, ss 2/20 |

(*Continued*)

Table 1. (Continued)

| Test format | Diagnostic reagent | RS# | Record | par act 1* | par act mult* | par inact 1* | par inact mult* | expar sub* | expar ven* | expar (NS)* | par +expar act* | CG** | OIG** | CG + OIG** | Cross-reactivity |
|---|---|---|---|---|---|---|---|---|---|---|---|---|---|---|---|
|  | cystic fluid | CT, s/b | Chang 1988 [63] | - | 88.2 (15/17) | - | - | 100.0 (1/1) | 100.0 (5/5) | - | 100.0 (11/11) | NS (NS/50) | NS (NS/4) | 90.7 (49/54) | NS |
|  | [cystic fluid] | CT, s/b | Cho 1986 [64] | - | - | - | - | 75.0 (3/4) | - | - | - | 94.2 (49/52) | 93.6 (103/110) | 93.8 (152/162) | Tsag 2/18, sp 2/20, pg 1/56, cl 1/15, fs 1/1 |
|  | cyst fluid | CT, MRI | Ferrer 2005a [35] | - | - | - | 100.0 (31/31) | - | - | - | - | 100.0 (30/30) | 49.1 (28/57) | 66.7 (58/87) | ce 17/20, Ssm 5/13, fs 3/9, tc 4/15 |
|  | cyst fluid | CT, MRI | Ferrer 2005b [36] | - | - | - | 100.0 (20/20) | - | - | - | - | 100.0 (78/78) | 63.4 (90/142) | 76.4 (168/220) | ce 17/20, Ssm 5/13, fs 3/9, tc 4/15, Hmn 1/2, oc 3/4, anc 2/3, tr 2/5, as 7/7, ml 1/3, tp 0/15, Amh 2/11, Blh 2/2, Eln 0/2, gi 0/4, ch 3/8, Lsc 0/9, hep 0/5, ms 0/4, cm 0/1 |
|  | cyst fluid | CT, MRI | Ferrer 2007a [37] | - | - | - | 100.0 (31/31) | - | - | - | - | 100.0 (30/30) | 49.1 (28/57) | 66.7 (58/87) | ce 17/20, Ssm 5/13, fs 3/9, tc 4/15 |
|  | cysticercal fluid | CT, MRI | Fleury 2007 [70] | - | - | - | - | - | - | 96.5 (28/29) | - | - | - | - | - |
|  | vesicular fluid | NS | Hernández 2019 [67] | - | - | - | - | - | - | 100.0 (29/29) | - | 85.4 (35/41) | - | - | - |
|  | NovaLisa Taenia solium IgG kit Ag (NovaTec Immundiagnostica GmbH, Dietzenbach, Germany) | NS | Hernández 2019 [67] | - | - | - | - | - | - | 82.8 (24/29) | - | 97.1 (33/34) | - | - | - |
|  | crude cyst fluid | CT, MRI | Lee 2005a [62] | - | - | - | 60.0 (12/20) | - | - | - | - | 100.0 (50/50) | 77.1 (145/188) | 81.9 (195/238) | Tsot 4/15, Tsag 2/15, Tas 1/10, ae 5/8, ce 23/50, sp 3/30, pg 2/30, cl 3/30 |
|  | 120kDa (CF) | CT, MRI | Lee 2005a [62] | - | - | - | 35.0 (7/20) | - | - | - | - | 100.0 (50/50) | 97.9 (184/188) | 98.3 (234/238) | Tsot 0/15, Tsag 0/15, Tas 0/10 ae 1/8, ce 3/50, sp 1/30, pg 0/30, cl 0/30 |
|  | crude cyst fluid | CT, MRI | Lee 2005b [65] | - | - | - | 60.0 (6/10) | - | - | - | - | 100.0 (25/25) | 78.9 (90/114) | 82.7 (115/139) | sp 3/20, ae 2/5, ce 13/29, pg 3/30, cl 3/30 |
|  | IEF-purified cyst Ag (CF) | CT, MRI | Oommen 2004 [51] | 41.5 (17/41) | - | - | - | - | - | - | - | 84.6 (22/26) | - | - | - |
|  | cyst fluid | CT, MRI | Pappala 2017 [56] | - | - | - | - | - | 0.0 (0/1) | - | 37.1 (13/35) | NS (NS/200) | - | - | - |

(Continued)

**Table 1.** (Continued)

| Test format | Diagnostic reagent | RS# | Record | par act 1* | par act mult* | par inact 1* | par inact mult* | expar sub* | expar ven* | expar (NS)* | par +expar act* | CG** | OIG** | CG + OIG** | Cross-reactivity |
|---|---|---|---|---|---|---|---|---|---|---|---|---|---|---|---|
| | low-molecular weight Ag (CF) | MRI | Sako 2015 [38] | 75.0 (3/4) | 100.0 (15/15) | - | - | 85.7 (6/7) | 50.0 (1/2) | - | 100.0 (4/4) | 100.0 (24/24) | - | - | - |
| | chimera 120kDa + 150 kDa (CF) | CT, MRI | Bae 2008 [74] | - | - | - | 27.5 (11/40) | - | - | - | - | 100.0 (60/60) | 93.8 (198/211) | 95.2 (258/271) | Tsot 1/14, Tsag 2/20, Tas 1/25, ae 3/15, ce 2/37, sp 2/20, pg 0/20, cl 1/20, fs 1/20, ss 0/20 |
| *T. solium* somatic Ag: Membrane | | | | | | | | | | | | | | | |
| | membrane | CT, MRI | Arruda 2005 [34] | - | - | - | - | - | - | - | 0.0 (0/1) | NS (NS/48) | NS (NS/32) | 96.2 (77/80) | sy NS/6, tp NS/6, Lsd NS/3, Ssm NS/4, Ssh NS/2, cm NS/2, im NS/3, hepA NS/2, hepB NS/4 |
| | cyst wall | CT, MRI | Pappala 2017 [56] | - | - | - | - | - | 100.0 (1/1) | - | 40.0 (14/35) | NS (NS/200) | - | - | - |
| | membrane | CT, s/b | Rosas 1986 [77] | - | - | - | - | 53.2 (25/47) | 33.3 (1/3) | 42.9 (3/7)^Δ | 52.1 (25/48) | 69.2 (385/556) | - | - | - |
| *T. solium* somatic Ag: Scolex | | | | | | | | | | | | | | | |
| | scolex | CT, MRI | Arruda 2005 [34] | - | - | - | - | - | - | - | 0.0 (0/1) | NS (NS/48) | NS (NS/32) | 96.2 (77/80) | sy NS/6, tp NS/6, Lsd NS/3, Ssm NS/4, Ssh NS/2, cm NS/2, im NS/3, hepA NS/2, hepB NS/4 |
| | protoscolex | CT, MRI | Pappala 2017 [56] | - | - | - | - | - | - | - | 42.9 (15/35) | NS (NS/200) | - | - | - |
| *T. solium* ESP | | | | | | | | | | | | | | | |
| | excretory secretory Ag | CT, MRI | Atluri 2009b [55] | 32.1 (35/109) | - | - | - | - | - | - | - | NS (NS/85) | NS (NS/40) | 76.8 (96/125) | NS |
| *T. solium* LLGP | | | | | | | | | | | | | | | |
| | LLGP cyst Ag | CT, MRI | Prabhakaran 2004 [52] | 80.4 (86/107) | - | - | - | - | - | - | - | 94.0 (94/100) | 100.0 (7/7) | 94.4 (101/107) | fl 0/2, tp 0/3, ra 0/2 |
| | non-solubilized LLGP | CT, MRI | Oommen 2004 [51] | 80.4 (86/107) | - | - | - | - | - | - | - | 94.0 (94/100) | - | - | - |
| | urea-induced conformed LLGP | CT, MRI | Prabhakaran 2007 [54] | 41.7 (25/60) | - | - | - | - | - | - | - | 100.0 (40/40) | - | 100.0 (40/40) | - |

*(Continued)*

**Table 1.** (Continued)

| Test format | Diagnostic reagent | RS# | Record | par act 1* | par act mult* | par inact 1* | par inact mult* | expar sub* | expar ven* | expar (NS)* | par +expar act* | CG** | OIG** | CG + OIG** | Cross-reactivity |
|---|---|---|---|---|---|---|---|---|---|---|---|---|---|---|---|
| *T. crassiceps* somatic Ag | | | | | | | | | | | | | | | |
| | vesicular fluid | NS | Hernández 2019 [67] | - | - | - | - | - | - | 96.6 (28/29) | - | 95.1 (39/41) | - | - | - |
| *T. solium* recombinant Ag | | | | | | | | | | | | | | | |
| | r sHSP35.6 | CT, MRI | Ferrer 2005a [35] | - | - | - | 71.0 (22/31) | - | - | - | - | 100.0 (30/30) | 82.5 (47/57) | 88.5 (77/87) | ce 2/20, Ssm 3/13, fs 2/9, tc 3/15 |
| | r Ts8B1 (ESP) | CT, MRI | Ferrer 2007a [37] | - | - | - | 58.1 (18/31) | - | - | - | - | 100.0 (30/30) | 89.5 (51/57) | 93.1 (81/87) | ce 3/20, Ssm 1/13, fs 0/9, tc 2/15 |
| | r Ts8B2 (ESP) | CT, MRI | Ferrer 2007a [37] | - | - | - | 67.7 (21/31) | - | - | - | - | 100.0 (30/30) | 89.5 (51/57) | 93.1 (81/87) | ce 2/20, Ssm 2/13, fs 0/9, tc 2/15 |
| | r Ts8B3 (ESP) | CT, MRI | Ferrer 2007a [37] | - | - | - | 16.1 (5/31) | - | - | - | - | 100.0 (30/30) | 78.9 (45/57) | 86.2 (75/87) | ce 2/20, Ssm 3/13, fs 4/9, tc 3/15 |
| | rTs8B2-His (ESP) | CT, MRI | Ferrer 2009 [39] | - | - | - | 67.7 (21/31) | - | - | - | - | NS (NS/30) | NS (NS/57) | 93.1 (81/87) | ce NS/20, Ssm NS/13, fs NS/9, tc NS/15 |
| | rTs8B2-Bac (ESP) | CT, MRI | Ferrer 2009 [39] | - | - | - | 71.0 (22/31) | - | - | - | - | NS (NS/30) | NS (NS/57) | 95.4 (83/87) | ce NS/20, Ssm NS/13, fs NS/9, tc NS/15 |
| | r Ts8B2 (ESP) | CT, MRI | Ferrer 2012 [40] | - | - | - | 68.6 (35/51) | - | - | - | - | 100.0 (30/30) | 88.9 (56/63) | 92.5 (86/93) | ce 2/20, Ssm 2/13, fs 0/9, tc 2/15, oc 0/4, Hmn 1/2 |
| | r Ts8B2-NT (ESP) | CT, MRI | Ferrer 2012 [40] | - | - | - | 68.6 (35/51) | - | - | - | - | 100.0 (30/30) | 96.8 (61/63) | 97.8 (91/93) | ce 1/20, Ssm 1/13, fs 0/9, tc 0/15, oc 0/4, Hmn 0/2 |
| | r Ts8B2-CT (ESP) | CT, MRI | Ferrer 2012 [40] | - | - | - | 68.6 (35/51) | - | - | - | - | 100.0 (30/30) | 96.8 (61/63) | 97.8 (91/93) | ce 1/20, Ssm 1/13, fs 0/9, tc 0/15, oc 0/4, Hmn 0/2 |
| *T. solium* synthetic Ag | | | | | | | | | | | | | | | |
| | s Ts8B2-1 (ESP) | CT, MRI | Ferrer 2012 [40] | - | - | - | 3.9 (2/51) | - | - | - | - | 100.0 (30/30) | 100.0 (63/63) | 100.0 (93/93) | ce 0/20, Ssm 0/13, fs 0/9, tc 0/15, oc 0/4, Hmn 0/2 |
| | s Ts8B2-2 (ESP) | CT, MRI | Ferrer 2012 [40] | - | - | - | 0.0 (0/51) | - | - | - | - | 100.0 (30/30) | 100.0 (63/63) | 100.0 (93/93) | ce 0/20, Ssm 0/13, fs 0/9, tc 0/15, oc 0/4, Hmn 0/2 |
| | s Ts8B2-3 (ESP) | CT, MRI | Ferrer 2012 [40] | - | - | - | 5.9 (3/51) | - | - | - | - | 100.0 (30/30) | 100.0 (63/63) | 100.0 (93/93) | ce 0/20, Ssm 0/13, fs 0/9, tc 0/15, oc 0/4, Hmn 0/2 |

*(Continued)*

**Table 1.** (Continued)

| Test format | Diagnostic reagent | RS# | Record | par act 1* | par act mult* | par inact 1* | par inact mult* | expar sub* | expar ven* | expar (NS)* | par +expar act* | CG** | OIG** | CG + OIG** | Cross-reactivity |
|---|---|---|---|---|---|---|---|---|---|---|---|---|---|---|---|
| | s Ts8B2-4 (ESP) | CT, MRI | Ferrer 2012 [40] | - | - | - | 13.7 (7/51) | - | - | - | - | 100.0 (30/30) | 95.2 (60/63) | 96.7 (90/93) | ce 1/20, Ssm 1/13, fs 0/9, tc 1/15, oc 0/4, Hmn 0/2 |
| | s Ts8B2-5 (ESP) | CT, MRI | Ferrer 2012 [40] | - | - | - | 2.0 (1/51) | - | - | - | - | 100.0 (30/30) | 100.0 (63/63) | 100.0 (93/93) | ce 0/20, Ssm 0/13, fs 0/9, tc 0/15, oc 0/4, Hmn 0/2 |
| | s Ts8B2-6 (ESP) | CT, MRI | Ferrer 2012 [40] | - | - | - | 9.8 (5/51) | - | - | - | - | 100.0 (30/30) | 96.8 (61/63) | 97.8 (91/93) | ce 1/20, Ssm 1/13, fs 0/9, tc 0/15, oc 0/4, Hmn 0/2 |
| | CyDA-1 (8kDa) | CT, MRI | Ferrer 2012 [40] | - | - | - | 0.0 (0/51) | - | - | - | - | 100.0 (30/30) | 100.0 (63/63) | 100.0 (93/93) | ce 0/20, Ssm 0/13, fs 0/9, tc 0/15, oc 0/4, Hmn 0/2 |
| | CyDA-2 (8kDa) | CT, MRI | Ferrer 2012 [40] | - | - | - | 0.0 (0/51) | - | - | - | - | 100.0 (30/30) | 100.0 (63/63) | 100.0 (93/93) | ce 0/20, Ssm 0/13, fs 0/9, tc 0/15, oc 0/4, Hmn 0/2 |
| | CyDA-3 (8kDa) | CT, MRI | Ferrer 2012 [40] | - | - | - | 0.0 (0/51) | - | - | - | - | 100.0 (30/30) | 100.0 (63/63) | 100.0 (93/93) | ce 0/20, Ssm 0/13, fs 0/9, tc 0/15, oc 0/4, Hmn 0/2 |
| | CyDA-4 (8kDa) | CT, MRI | Ferrer 2012 [40] | - | - | - | 2.0 (1/51) | - | - | - | - | 100.0 (30/30) | 100.0 (63/63) | 100.0 (93/93) | ce 0/20, Ssm 0/13, fs 0/9, tc 0/15, oc 0/4, Hmn 0/2 |
| | CyDA-5 (8kDa) | CT, MRI | Ferrer 2012 [40] | - | - | - | 5.9 (3/51) | - | - | - | - | 93.3 (28/30) | 84.1 (53/63) | 87.1 (81/93) | ce 3/20, Ssm 2/13, fs 1/9, tc 2/15, oc 1/4, Hmn 1/2 |
| | CyDA-6 (8kDa) | CT, MRI | Ferrer 2012 [40] | - | - | - | 0.0 (0/51) | - | - | - | - | 100.0 (30/30) | 100.0 (63/63) | 100.0 (93/93) | ce 0/20, Ssm 0/13, fs 0/9, tc 0/15, oc 0/4, Hmn 0/2 |
| | *T. saginata* synthetic Ag | | | | | | | | | | | | | | |
| | s HP6-3 + Ts45W-1 + Ts45W-5 | CT, MRI | Ferrer 2005b [36] | - | - | - | 85.0 (17/20) | - | - | - | - | 100.0 (48/48) | 83.8 (119/142) | 87.9 (167/190) | ce NS, Ssm NS, fs NS, tc NS, Hmn NS, on NS, anc NS, tr NS, as NS, ml NS, tp NS, Amh NS, Blh NS, Eln NS, gi NS, ch NS, Lsc NS, hep NS, ms NS, cm NS |
| | s HP6-3 + Ts45W-1 + Ts45W-5 | CT, MRI | Ferrer 2007b [41] | - | - | - | 51.6 (16/31) | - | - | - | - | 100.0 (30/30) | 78.9 (45/57) | 86.2 (75/87) | ce 6/20, Ssm 2/13, fs 2/9, tc 2/15 |
| | s HP6-GST | CT, MRI | Ferrer 2007b [41] | - | - | - | 29.0 (9/31) | - | - | - | - | 100.0 (30/30) | 89.5 (51/57) | 93.1 (81/87) | ce 2/20, Ssm 2/13, fs 0/9, tc 2/15 |
| | s HP6-Bac | CT, MRI | Ferrer 2007b [41] | - | - | - | 64.5 (20/31) | - | - | - | - | 100.0 (30/30) | 93.0 (53/57) | 95.4 (83/87) | ce 1/20, Ssm 1/13, fs 0/9, tc 2/15 |

(*Continued*)

**Table 1.** (Continued)

| Test format | Diagnostic reagent | RS# | Record | par act 1* | par act mult* | par inact 1* | par inact mult* | expar sub* | expar ven* | expar (NS)* | par +expar act* | CG** | OIG** | CG + OIG** | Cross-reactivity |
|---|---|---|---|---|---|---|---|---|---|---|---|---|---|---|---|
| | s HP6-3 | CT, MRI | Ferrer 2007b [41] | - | - | - | 41.9 (13/31) | - | - | - | - | 100.0 (30/30) | 82.5 (47/57) | 88.5 (77/87) | ce 6/20, Ssm 2/13, fs 2/9, tc 0/15 |
| **DOT-ELISA** | *T. solium* somatic Ag | | | | | | | | | | | | | | |
| | 53/25kDa Ag (CF) | CT, MRI | Piña 2011 [42] | 35.3 (18/51) | 77.3 (43/55) | - | - | - | - | 97.2 (52/54) | - | 100.0 (96/96) | 98.9 (92/93) | 99.5 (188/189) | Asl 0/2, Env 0/8, Hmn 0/27, Dpp 0/2, Sts 0/12, Tsag 0/10, Trt 0/6, hw 0/8, ce 1/18 |
| | *T. taeniaeformis* somatic Ag | | | | | | | | | | | | | | |
| | membrane | CT, MRI | Shukla 2008 [57] | 83.3 (30/36) | 100.0 (14/14) | - | - | - | - | | | 73.3 (22/30) | 75.0 (15/20) | 74.0 (37/50) | Mbt 5/20 |
| **POC LFA** | *T. solium* recombinant Ag | | | | | | | | | | | | | | |
| | rT24H | CT | Stelzle 2024 [20] | - | - | 33.3 (1/3) | 85.0 (17/20)†4 | - | - | - | - | 91.0 (NS) | - | - | - |
| | rT24H | CT | Zulu 2024 [27] | - | - | 72.7 (8/11) | 73.7 (14/19)†5 | - | - | - | - | 88.0 (NS) | - | - | - |
| **MAPIA** | *T. solium* recombinant + synthetic Ag | | | | | | | | | | | | | | |
| | rGP50 + rT24H + sTs14 | CT, MRI | Toribio 2023a [31] | 100.0 (29/29) | 100.0 (9/9) | - | - | 100.0 (40/40) | - | - | - | 98.5 (67/68) | - | - | - |
| **Serological antigen detection** | | | | | | | | | | | | | | | |
| **ag ELISA** | mAb: TsW8/TsW5 | | | | | | | | | | | | | | |
| | TsW8/TsW5 | CT, MRI | Castillo 2023 [43] | - | - | - | - | 97.8 (47/48) | - | - | - | - | - | - | - |
| | mAb: B158/B60 | | | | | | | | | | | | | | |
| | test kit mAb B158/B60 (apDia, Turnhout, Belgium) | CT, MRI | Castillo 2023 [43] | - | - | - | - | 97.8 (47/48) | - | - | - | - | - | - | - |
| | B158/B60 | CT | Gabriël 2012 [71] | - | 100.0 (6/6)†6 | - | 36.4 (4/11)†7 | - | - | - | - | 85.7 (42/49) | - | - | - |
| | B158/B60 | CT, MRI | Oommen 2004 [51] | 10.0 (7/70) | - | - | - | - | - | - | - | 95.0 (95/100) | - | - | - |

*(Continued)*

**Table 1.** (Continued)

| Test format | Diagnostic reagent | RS# | Record | par act 1* | par act mult* | par inact 1* | par inact mult* | expar sub* | expar ven* | expar (NS)* | par +expar act* | CG** | OIG** | CG + OIG** | Cross-reactivity |
|---|---|---|---|---|---|---|---|---|---|---|---|---|---|---|---|
| | B158/B60 | CT, MRI | Rodriguez 2009 [30] | - | - | - | - | 100.0 (25/25) | 77.8 (7/9) | - | - | - | - | - | - |
| | B158/B60 | CT | Stelzle 2024 [20] | - | - | 33.3 (1/3) | 50.0 (10/20)†8 | - | - | - | - | 97.0 (NS) | - | - | - |
| | B158/B60 | CT, MRI | Zea-Vera 2013 [32] | - | - | - | - | - | 100.0 (1/1) | - | - | - | - | - | - |
| | B158/B60 | CT | Zulu 2024 [27] | - | 75.0 (6/8)†9 | 27.3 (3/11) | 15.8 (3/19)†10 | - | - | - | - | 82.0 (NS) | - | - | - |
| | mAb: HP10 | | | | | | | | | | | | | | |
| | HP10 | CT, MRI | Ferrer 2005a [35] | - | - | - | 12.9 (4/31) | - | - | - | - | 100.0 (30/30) | 96.5 (55/57) | 97.7 (85/87) | ce 1/20, Ssm 0/13, fs 0/9, tc 1/15 |
| | HP10 | CT, MRI | Ferrer 2005b [36] | - | - | - | 10.0 (2/20) | - | - | - | - | 100.0 (48/48) | 96.5 (137/142) | 97.4 (185/190) | ce 1/20, Ssm 0/13, fs 0/9, tc 1/15, Hmn 0/2, on 0/4, anc 0/3, tr 0/5, as 2/7, ml 0/3, tp 0/15, Amh 0/11, Blh 0/2, Eln 0/2, gi 0/4, ch 1/8, Lsc 0/9, hep 0/5, ms 0/4, cm 0/1 |
| | HP10 | CT, MRI | Ferrer 2007a [37] | - | - | - | 12.9 (4/31) | - | - | - | - | 100.0 (30/30) | 96.5 (55/57) | 97.7 (85/87) | ce 1/20, Ssm 0/13, fs 0/9, tc 1/15 |
| | HP10 | CT, MRI | Fleury 2007 [70] | - | - | - | - | - | - | 84.8 (39/46) | - | NS (NS/115) | NS (NS/36) | 94.0 (142/151) | Amc NS, Gil NS, Asl NS, Amh NS |
| | HP10 | CT | García 2002 [44] | - | - | - | - | - | - | 37.5 (6/16) | 61.5 (8/13) | - | - | - | - |
| | HP10 | NS | Hernández 2019 [67] | - | - | - | - | - | - | 89.7 (26/29) | - | 94.9 (39/41) | - | - | - |
| | HP10 | CT, MRI | Parkhouse 2018 [45] | - | - | - | - | - | - | 80.0 (4/5) | 77.8 (7/9) | - | - | - | - |
| **Urine-based antibody detection** | | | | | | | | | | | | | | | |
| ab ELISA | T. solium somatic Ag: WCE | | | | | | | | | | | | | | |
| | crude soluble extract | CT, MRI | Atluri 2009b [55] | 42.2 (46/109) | - | - | - | - | - | - | - | NS (NS/85) | NS (NS/40) | 66.4 (83/125) | ml NS/10, tp NS/7, hd NS/9, am NS/12, as NS/2 |
| | low-molecular mass 10-30kDa | CT, MRI | Atluri 2009b [55] | 40.4 (44/109) | - | - | - | - | - | - | - | NS (NS/85) | NS (NS/40) | 58.4 (73/125) | ml NS/10, tp NS/7, hd NS/9, am NS/12, as NS/2 |

(*Continued*)

**Table 1.** (Continued)

| Test format | Diagnostic reagent | RS# | Record | par act 1* | par act mult* | par inact 1* | par inact mult* | expar sub* | expar ven* | expar (NS)* | par +expar act* | CG** | OIG** | CG + OIG** | Cross-reactivity |
|---|---|---|---|---|---|---|---|---|---|---|---|---|---|---|---|
| | *T. solium* ESP | | | | | | | | | | | | | | |
| | excretory secretory Ag | CT, MRI | Atluri 2009b [55] | 44.0 (48/109) | - | - | - | - | - | - | - | NS (NS/85) | NS (NS/40) | 65.2 (82/125) | ml NS/10, tp NS/7, hd NS/9, am NS/12, as NS/2 |
| **Urine-based antigen detection** | | | | | | | | | | | | | | | |
| **ag ELISA** | mAb: B158/B60 | | | | | | | | | | | | | | |
| | B158/B60 | CT, MRI | Castillo 2009 [17] | 62.5 (5/8) | 96.2 (25/26) | - | - | - | - | - | - | 100.0 (24/24) | - | - | - |
| **POC dipstick** | mAb: TsW8/TsW5 | | | | | | | | | | | | | | |
| | TsW8/TsW5 | NS | Toribio 2023b [46] | - | - | - | - | 96.7 (29/30) | - | - | - | 100.0 (10/10) | - | - | - |

RS = reference standard, Par act 1 = parenchymal active single cysts, par act mult = parenchymal active multiple cysts, par inact 1 = parenchymal inactive single cysts, par inact mult = parenchymal inactive multiple cysts, expar sub = extraparenchymal subarachnoid cysts, expar ven = extraparenchymal ventricular cysts, expar (NS) = extraparenchymal cysts without localization specified, par +expar act = parenchymal and extraparenchymal active cysts, CG = control group, OIG = other infections group, CG+OIG = control group and other infections group; # options: MRI: magnetic resonance imaging, CT: computed tomography, imaging: not further specified, NS: not specified but data on cyst localization and phase available, s/b: surgery/biopsy

\* Sensitivity percentage (true positives / (true positives + false negatives))

\*\* Specificity percentage (true negatives / (true negatives + false positives))

Ag = antigen, WCE = whole cyst extract, CF = cyst fluid, ESP = excretory-secretory protein, LLGP = lentil lectin-bound glycoprotein, (m)Ab = (monoclonal) antibody

† Specified sensitivity data for multiple cysts

†1 2–5 cysts: 0.0 (0/6), >5 cysts: 35.7 (5/14)

†2 2–5 cysts: 16.7 (1/6), >5 cysts: 64.3 (9/14)

†3 2–5 cysts: 50.0 (6/12), >5 cysts: 28.6 (2/7)

†4 2–5 cysts: 83.3 (5/6), >5 cysts: 85.7 (12/14)

†5 2–5 cysts: 83.3 (10/12), >5 cysts: 57.1 (4/7)

†6 >5 cysts: 100.0 (6/6)

†7 2–5 cysts: 50.0 (3/6), >5 cysts: 20.0 (1/5)

†8 2–5 cysts: 33.3 (2/6), >5 cysts: 57.1 (8/14)

†9 2–5 cysts: 75.0 (3/4), >5 cysts: 75.0 (3/4)

†10 2–5 cysts: 16.7 (2/12), >5 cysts: 14.3 (1/7)

Δ Extraparenchymal inactive cysts

ml = malaria, tp = toxoplasmosis, hd = hydatidosis, am = amoebiasis, as = ascariasis, Tsot = *T. solium* taeniasis, Tas = *T. asiatica* taeniasis, Tsag = *T. saginata* taeniasis, ae = alveolar echinococcosis, ce = cystic echinococcosis, sp = sparganosis, pg = paragonimiasis, cl = clonorchiasis, fs = fascioliasis, ss = schistosomiasis, fl = filariasis, ra = rheumatoid arthritis, dp = diphyllobothriasis, Ssj = *Schistosoma japonicum*, an = anisakiasis, tr = trichuriasis, Hmn = *Hymenolepis nana*, Dpp = *Diphyllobothrium pacificum*, Asl = *Ascaris lumbricoides*, Env = *Enterobius vermicularis*, Trt = *Trichuris trichiura*, anc = ancylostomiasis, Sts = *Strongyloides stercoralis*, sy = syphilis, Lsd = *Leishmania donovani*, Ssm = *Schistosoma haematobium*, cm = cytomegalovirus, im = infectious mononucleosis, hepA = hepatitis A, hepB = hepatitis B, tc = toxocariasis, oc = onchocerciasis, Amh = *Entamoeba histolytica*, Blh = *Blastocystis hominis*, Eln = *Endolimax nana*, gi = giardiasis, ch = Chagas' disease, Lsc = *Leishmania chagasi*, hep = hepatitis, ms = measles, hw = hookworm, Mbt = *Mycobacterium tuberculosis*, Amc = *Entamoeba coli*, Gil = *Giardia lamblia*.

Table 2. Test sensitivity and specificity of serological antibody- and antigen-detecting tests, included with low confidence, categorized per test format and diagnostic reagent.

| Test format | Diagnostic reagent | RS# | Record | par act 1* | par act mult* | expar sub* | expar (NS)* | CG** | OIG** | CG +OIG** | Cross-reactivity |
|---|---|---|---|---|---|---|---|---|---|---|---|
| **Serological antibody detection** | | | | | | | | | | | |
| **Western Blot** | *T. solium* LLGP | | | | | | | | | | |
| | LLGP cyst Ag (CDC) | CT | Dermauw 2018 [72] | 66.7 (2/3) | 80.0 (4/5) | - | - | 98.2 (111/113) | - | - | - |
| | Qualicode Cysticercosis EITB kit Ag (Immunetics Inc, Cambridge, MA) | CT | Dermauw 2018 [72] | 100.0 (3/3) | 100.0 (3/3) | - | - | 94.0 (63/67) | - | - | - |
| | LLGP cyst Ag (CDC) | CT | Palacio 1998 [29] | 41.2 (7/17) | 81.8 (18/22) | - | - | 98.0 (350/357) | - | - | - |
| | LLGP cyst Ag | CT | Stelzle 2024 [20] | - | 67.9 (19/28)†1 | - | - | 97.0 (NS) | - | - | - |
| | *T. solium* recombinant Ag | | | | | | | | | | |
| | rT24H | CT | Dermauw 2018 [72] | 66.7 (2/3) | 80.0 (4/5) | - | - | 98.2 (111/113) | - | - | - |
| | rT24H | CT | Stelzle 2024 [20] | - | 100.0 (28/28) | - | - | 98.0 (NS) | - | - | - |
| **ab ELISA** | *T. solium* somatic Ag: WCE | | | | | | | | | | |
| | crude soluble extract | CT | Mandal 2006 [58] | 87.0 (60/69) | 100.0 (11/11) | - | - | 90.0 (54/60) | 67.5 (27/40) | 81.0 (81/100) | Mbt 8/20, ce 3/5, tp 1/5, ml 1/5, am 0/5 |
| | low-molecular mass 20-24kDa | CT | Mandal 2008 [59] | 84.1 (58/69) | 100.0 (11/11) | - | - | 100.0 (60/60) | 100.0 (40/40) | 100.0 (100/100) | Mbt 0/20, ce 0/5, tp 0/5, ml 0/5, am 0/5 |
| | whole cysticerci | CT, s/b | Mohammad 1984 [69] | - | - | 87.5 (7/8) | 100.0 (3/3) | 100.0 (19/19) | - | - | - |
| | *T. solium* somatic Ag: CF | | | | | | | | | | |
| | [cystic fluid] | CT, s/b | Cho 1986 [64] | - | 82.2 (37/45) | - | - | 94.2 (49/52) | 93.6 (103/110) | 93.8 (152/162) | Tsag 2/18, sp 2/20, pg 1/56, cl 1/15, fs 1/1 |
| **DOT-ELISA** | *T. solium* somatic Ag | | | | | | | | | | |
| | Tso crude extract | imaging | Agudelo 2005 [47] | - | 91.1 (41/45) | - | - | 100.0 (37/37) | 100.0 (43/43) | 100.0 (80/80) | Tsot 0/2, Tpg 0/4, Sts 0/15, df 0/1, multiple 0/2, Ocv 0/2, br 0/1, pl 0/1, Amh 0/15 |
| | Tso crude soluble extract | CT | Mandal 2006 [58] | 87.0 (60/69) | 100.0 (11/11) | - | - | 83.3 (50/60) | 57.5 (23/40) | 73.0 (73/100) | Mbt 10/20, ce 3/5, tp 4/5, ml 0/5, am 0/5 |
| | low-molecular mass 20-24kDa | CT | Mandal 2008 [59] | 84.1 (58/69) | 100.0 (11/11) | - | - | 100.0 (60/60) | 95.0 (38/40) | 98.0 (98/100) | Mbt 0/20, ce 2/5, tp 0/5, ml 0/5, am 0/5 |

(*Continued*)

**Table 2.** (Continued)

| Test format | Diagnostic reagent | RS# | Record | par act 1* | par act mult* | expar sub* | expar (NS)* | CG** | OIG** | CG + OIG** | Cross-reactivity |
|---|---|---|---|---|---|---|---|---|---|---|---|
| **POC LFA** | *T. solium* recombinant Ag | | | | | | | | | | |
| | rT24H | CT | Stelzle 2024 [20] | - | 100.0 (28/28) | - | - | 91.0 (NS) | - | - | - |
| **Serological antigen detection** | | | | | | | | | | | |
| **ag ELISA** | mAb: B158/B60 | | | | | | | | | | |
| | B158/B60 | CT | Stelzle 2024 [20] | - | 100.0 (28/28) | - | - | 97.0 (NS) | - | - | - |

RS = reference standard, Par act 1 = parenchymal active single cysts, par act mult = parenchymal active multiple cysts, expar sub = extraparenchymal subarachnoid cysts, expar (NS) = extraparenchymal cysts without localization specified, CG = control group, OIG = other infections group, CG+OIG = control group and other infections group

# options: CT: computed tomography, s/b: surgery/biopsy, imaging: not further specified

\* Sensitivity percentage (true positives / (true positives + false negatives))

\*\* Specificity percentage (true negatives / (true negatives + false positives))

LLGP = lentil lectin-bound glycoprotein, Ag = antigen, WCE = whole cyst extract, CF = cyst fluid, (m)Ab = (monoclonal) antibody† Specified sensitivity data for multiple cysts

†1 2–5 cysts: 50.0 (1/2), >5 cysts: 69.2 (18/26); Mbt = *Mycobacterium tuberculosis*, ce = cystic echinococcosis, tp = toxoplasmosis, ml = malaria, am = amoebiasis, Tsag = *T. saginata* taeniasis, sp = sparganosis, pg = paragonimiasis, cl = clonorchiasis, fs = fascioliasis, Tsot = *T. solium* taeniasis, Tpg = *Toxoplasma gondii*, Sts = *Strongyloides stercoralis*, df = dirofilariasis, Ocv = *Onchocerca volvulus*, br = brucellosis, pl = plasmodiasis, Amh = *Entamoeba histolytica*.

Tests using diagnostic reagents derived from *Taenia crassiceps* [67], *Taenia saginata* [36,41], and *Taenia taeniaeformis* [57], constitute a possible alternative to the commonly used *T. solium* metacestode antigen. Also displayed in Table 1 are five studies evaluating promising rapid tests or POC tests, two describing DOT-ELISAs in phase of test development, using either *T. solium* somatic antigen as diagnostic reagent (sensitivity parenchymal active multiple cysts 77.3% (43/55), extraparenchymal cysts 97.2% (52/54), overall specificity 99.5% (188/189)) [42], or using *T. taeniaeformis* somatic antigen as reagent (sensitivity parenchymal active single cysts 83.3% (30/36), parenchymal active multiple cysts 100.0% (14/14), overall specificity 74.0% (37/50)) [57]. One describes a urine-based dipstick assay with laboratory-based evaluation of samples from subarachnoid NCC patients (sensitivity 96.7% (29/30), specificity control group 100.0% (10/10)) [46]. However, all these tests require further evaluation in setting-specific population-based studies to contribute as screening tools in rural setting. Another two tests describe a lateral flow-assay with potential use as POC test, tested in a controlled hospital setting in Tanzania [20], and in a community-based setting in Zambia [27]. Although many data here were excluded for analysis due to insufficient characterization and sample size, these studies reported sensitivity values of 78.3% (18/23) and 73.3% (22/30) respectively for parenchymal inactive cysts, and control group specificity of 91.0% and 88.0%. Data included with low confidence can be found in Table 2. In the supplementary materials, an overview of sensitivity and specificity data of all 169 records and associated tests can be found (S2 Table).

## Discussion

The goal of this systematic review was to provide an overview of the existing (and in literature described) serological and urine-based immunological tests for diagnosis of NCC, with a main focus on evaluating test performance according to cyst localization, stage and number. An accurate and early disease detection is especially important in patients with active NCC. More specifically, early detection of active NCC via immunodiagnosis could shorten the diagnostic pathway between screening and final neuroimaging confirmation, and therefore benefit early initiation of treatment. In case of parenchymal active NCC, treatment with anthelmintics may induce degeneration and calcification of active cysts, leading to cyst resolution [7,71]. As a result, clinical symptoms may subside or even cease completely, significantly improving the quality of life. Also, immunodiagnosis could be potentially beneficial in monitoring treatment efficacy, as the change in cyst stage is associated with a drop in antigen levels, indicating treatment success [32,78]. In the case of extraparenchymal active NCC, an early detection is paramount for patient referral to specialized care and treatment, such as surgical cyst removal to avoid life-threatening disease development [7].

The heterogeneity of this pleiomorphic disease complicates the interpretation of results and the usability of tests. Host immunological responses are driven by cyst localization, stage and number. Therefore, in order to perform an appropriate evaluation of test performance, well-characterized data is required. This review revealed that data on patient characteristics was scarce in many records, with ambiguous or no cyst specifications regarding localization or stage. To address this issue and exploit as much data as possible, a standardized method was developed for classification of cyst characteristics, based on certainty criteria for confidence scaling (Fig 1). Of the 169 records selected for data collection, 53 studies assessing 123 tests were ultimately included with high and/or low confidence. This indicates that even with standardized data extraction, supplied data on cyst characteristics is insufficient in the majority of published literature. Unequivocal data on used test format and diagnostic reagent is also paramount to impede exclusion and enable test comparison in this review. Although many records indicated the used reagent and test characterization, specified data is needed on reagent

procurement and preparation/synthesis method, on the used test methodology and threshold determination, and on the detected analyte. Also, antigens used for coating require specification on the used strain/species of the parasite, as different results could be expected whether or not using the indigenous local strain/species [79]. While all records provided test format data, some lacked details on the used diagnostic reagent, requiring further research in other data sources or contacting the author for clarification. Accessibility to these immunodiagnostic tests and reagents is another limiting factor. Test comparison is further complicated by patient population heterogeneity. Patients enrolled in a hospital-based setting are expected to have more severe symptoms, associated with higher antigen/antibody levels, compared to individuals enrolled in community-based studies for screening purposes [78]. Some studies also performed preliminary testing on the patient population, only including these patients for index testing who had a previous positive test result. Further bias is induced depending on the used test to define patient recruitment, e.g. antigen ELISAs -which were specifically developed to detect active cysts [80] -will naturally detect more cases with active multiple cysts. Similar for reference standards, e.g. CT scans will bias towards selection of patients with parenchymal calcified cysts, whereas MRI scans are superior for detection of active parenchymal and extraparenchymal NCC [7].

Possible false-positive and false-negative test results must also be taken into account. Reference standards, as defined, are likely to correctly identify the target disease, however are not completely reliable as gold standard diagnostic tools. A biopsy only guarantees definite diagnosis when subsequent pathological confirmation is done. Imaging scans can display typical pathognomonic lesions for NCC, but may also be limited to showing highly suggestive lesions or compatible lesions not clearly discernible from lesions caused by other conditions [11]. Therefore, in studies with both definitive and probable cases of NCC, as defined by Del Brutto criteria [11], data was collected in this review's NCC group only for cases with definitive NCC. For control group and other infections group individuals, the uncertainty of correct classification is even greater, as many individuals have not been confirmed via reference standard. The RoB assessment showed that, of 148 studies with a defined control group and/or other infections group, only 20% specifically mentions concerned individuals to be confirmed as NCC-negative via reference standard (Fig 3 Q4). When evaluating specificity and cross-reactivity in these groups, we must also consider the possibility of other (undiagnosed) infections, depending on regional endemicity of these infections. Similar antigenic components to *T. solium* cysticercosis could cause false-positive index test results [9]. We must always hold into account that test performance results are highly setting-specific, and can not be plainly extrapolated. Furthermore, immunodiagnosis can yield false-positives for NCC when cysticerci are localized outside the central nervous system, or in case of transient seropositives [21,81].

In addition to the STARD 2015 reporting guideline for diagnostic accuracy studies, aimed at enhancing reporting completeness and transparency, and allowing adequate assessment of study validity [82], we wish to propose reporting recommendations for diagnostic accuracy studies assessing immunodiagnosis of neurocysticercosis. This includes a minimal and recommended set of information to be reported on methodology and results, detailed in the supplementary materials (S3 Table).

Abovementioned findings demonstrate the difficulty with which accuracy results of immunodiagnostic tests can be interpreted and compared, even with essential data available. Also, the RoB assessment of this review demonstrates that many studies likely carry bias, either in patient selection, in index testing, reference standard testing, or in flow and timing. Reported test results should therefore always be interpreted attentively, including highly promising results. Nonetheless, it is possible to cautiously formulate initial findings based on results of this systematic review. For detection of parenchymal active multiple cysts, the antibody-

detecting LLGP Western Blot approaches the high accuracy standards, as do antigen-detecting test formats. Antibody-detecting ELISA results are difficult to compare due to the variety of diagnostic reagents used. While these tests showed overall higher sensitivity for multiple cysts than antigen-detecting tests, due to high amount of circulating antibody in comparison to circulating antigen levels, they cannot discriminate active from inactive infections, making them inadequate for post-treatment follow-up. In this scenario, patient follow-up via urine-based antigen-detection could form a minimally invasive monitoring technique. More studies are needed to unveil the actual usability of urine tests. Up till now, the detection of parenchymal active single cysts remains challenging. The newly developed MAPIA based on *T. solium* recombinant and synthetic antigens [31], seemingly exceeded sensitivity results of other test formats for single cyst detection. It also showed promising results for diagnosis of parenchymal active multiple and extraparenchymal NCC. We must hold into account that current study results are based on low sample sizes, with patients selected using the LLGP Western Blot as additional reference standard to CT or MRI.

Some test results are promising but were excluded from Tables 1 and 2 due to inadequate characterization or insufficient sample size. These results can be consulted in the supplementary S2 Table. For example, an immunochromatography-based POC LFA (iCysticercosis kit) showed preliminary sensitivity of 83.3% (15/18, of which only 3 samples could be characterized with high confidence as extraparenchymal subarachnoid) and specificity of 92.0% (150/163) [9]. Another POC LFA based on rT24H-Qdots, gave -as determined in this review- an NCC group sensitivity of 81.3% (91/112, all patients with parenchymal and/or extraparenchymal active single or multiple cysts) and a control group + other infections group specificity of 98.6% (150/152) [83]. The rT24H-POC LFA tested in a hospital-based setting in Tanzania [20] and a community-based setting in Zambia [27], with estimated sensitivities of 49% and 26% for all types of NCC determined via logistic regression analysis and generalized linear models respectively, showed promising sensitivities for parenchymal active lesions (>98% and >99%), although sample sizes were too small to draw significant conclusions. Regarding antigen-detection, an antigen ELISA based on monoclonal antibody HP10 proved especially interesting for identification of patients with extraparenchymal NCC [45]. For use in rural field settings, the same research group has developed a modified HP10-antigen LFA with 100.0% (4/4) sensitivity for tested samples from patients with extraparenchymal cysts, and 75.0% (6/8) sensitivity for tested samples from patients with both parenchymal and extraparenchymal cysts [84].

Our study had some important limitations. Some publication bias may have been introduced, as mainly published records were screened. The full-text screening, data collection and RoB assessment were performed by one author per language only due to time and resource limitations. Yet for data collection, a 10% quality control of English articles was completed. The QUADAS-2 RoB assessment was performed only assessing risk of bias and not applicability of eligible studies, as we believed our eligibility criteria were robust enough to exclude those articles not aligning with the review questions. To maintain a sufficient level of confidence in cyst characteristics data, confidence scaling was done according to authors' predeveloped criteria, resulting in exclusion of many records and corresponding tests. As such, accuracy data might have been skewed and this must be considered when consulting the results. Another key limitation is the variability in reference standards across included studies. While neuroimaging is commonly used, it can not be truly considered as a gold standard technique, potentially leading to misclassification errors. We did not evaluate if included studies applied methods (e.g. Bayesian latent class models) to address potential errors. Future reviews could consider this aspect to strengthen conclusions. Lastly, this review has highlighted that interpretation and comparison of test results was highly challenging due to data heterogeneity, impeding the

possibility to perform meta-analysis. Nevertheless, the narrative analysis effectively elucidates immunological test performance across clinically relevant scenarios.

## Conclusion

To assess the potential added value of immunological tests in diagnosis of NCC, unambiguous and complete data on test performance is necessary. This requires researchers to ensure adequate characterization of samples during the study process, and to report all relevant data in published records. In conducted studies, bias can be minimized by avoiding predetermined classification of patients, and by determination of the disease status by use of reliable reference standard(s). Index test methodologies should be clearly defined, as well as used threshold values or determination methods, to facilitate comparison of test results. Neurocysticercosis is a global disease of major concern, mostly endemic in resource-poor areas where neuroimaging is often not available. Immunodiagnostic tests can help to provide an early and adequate diagnosis in settings ranging from specialized hospitals to rural communities. New test formats are in constant development, however, have not been validated due to lack of adequate sample size and well-characterized samples. A striving for an elaborate and accessible biobank, acquirable through extensive international collaboration, should be prioritized in order to further develop immunological tests. Thus far, especially suitable tests formats for use in resource-poor areas lack sufficiently large-scale evaluations in the targeted field settings. Point-of-care tests require further development and testing in targeted settings. A clear view on test characteristics and performance can subsequently be reflected in revised WHO TPPs, with recommendations adapted to contextual test use.

## Supporting information

**S1 Checklist. Preferred Reporting Items for Systematic Reviews and Meta-Analyses (PRISMA) 2020 checklist.**
(DOCX)

**S1 Search Strategy. Search strategy for databases searched.**
(DOCX)

**S1 Protocol. Published PROSPERO protocol.**
(PDF)

**S1 Text. Adapted signalling questions for QUADAS-2 Risk of Bias assessment.**
(DOCX)

**S1 Table. Quality Assessment (QUADAS-2) summary of included studies: Risk of Bias.**
(DOCX)

**S2 Table. Test sensitivity and specificity of serological and urine-based antibody- and antigen-detecting tests, categorized by test format and specimen.**
(XLSX)

**S3 Table. Reporting recommendations for diagnostic accuracy studies regarding neurocysticercosis diagnostic tests.**
(XLSX)

**S4 Table. Third screening phase (full-text screening) excluded records with reason of exclusion.**
(DOCX)

**S5 Table. Confidence scaling and record selection procedure.**
(XLSX)

## Acknowledgments

This work was supported by the NeuroSolve consortium. Following NeuroSolve consortium members from various institutions are acknowledged:

1. University of Dar es Salaam, Tanzania: Bernard Ngowi, Mkunde Chachage
2. Sokoine University of Agriculture, Tanzania: Helena Ngowi, Ernatus Mkupasi
3. University of Zambia, Zambia: Kabemba Evans Mwape, Gideon Zulu
4. Ghent University, Belgium: Sarah Gabriël, Lisa Van Acker
5. R-Evolution Worldwide Impresa Sociale, Italy: Dario Scaramuzzi

## Author Contributions

**Conceptualization:** Lisa Van Acker, Brecht Devleesschauwer, Héctor H. Garcia, Sarah Gabriël.

**Data curation:** Lisa Van Acker, Luz Toribio, Mkunde Chachage, Hang Zeng.

**Formal analysis:** Lisa Van Acker, Luz Toribio, Mkunde Chachage.

**Funding acquisition:** Sarah Gabriël.

**Investigation:** Lisa Van Acker, Luz Toribio, Mkunde Chachage, Hang Zeng, Sarah Gabriël.

**Methodology:** Lisa Van Acker, Brecht Devleesschauwer, Héctor H. Garcia, Sarah Gabriël.

**Project administration:** Lisa Van Acker, Brecht Devleesschauwer, Sarah Gabriël.

**Supervision:** Brecht Devleesschauwer, Sarah Gabriël.

**Validation:** Lisa Van Acker, Luz Toribio, Mkunde Chachage, Sarah Gabriël.

**Visualization:** Lisa Van Acker, Sarah Gabriël.

**Writing – original draft:** Lisa Van Acker.

**Writing – review & editing:** Lisa Van Acker, Luz Toribio, Mkunde Chachage, Hang Zeng, Brecht Devleesschauwer, Héctor H. Garcia, Sarah Gabriël.

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
