## [Decision Letter · Decision Letter 0]

26 Sep 2024

Dear Ms. Van Acker,

Thank you very much for submitting your manuscript "Accuracy of immunological tests on serum and urine for diagnosis of *Taenia solium* neurocysticercosis: A systematic review" for consideration at PLOS Neglected Tropical Diseases. As with all papers reviewed by the journal, your manuscript was reviewed by members of the editorial board and by several independent reviewers. The reviewers appreciated the attention to an important topic. Based on the reviews, we are likely to accept this manuscript for publication, providing that you modify the manuscript according to the review recommendations. 

Sincerely,

Christine M. Budke

Academic Editor

Francesca Tamarozzi

Section Editor

Reviewer's Responses to Questions

**Key Review Criteria Required for Acceptance?**

**Methods**

-Are the objectives of the study clearly articulated with a clear testable hypothesis stated?

-Is the study design appropriate to address the stated objectives?

-Is the population clearly described and appropriate for the hypothesis being tested?

-Is the sample size sufficient to ensure adequate power to address the hypothesis being tested?

-Were correct statistical analysis used to support conclusions?

-Are there concerns about ethical or regulatory requirements being met?

Reviewer #1: Well formulated objectives of the study. Appropriate searching and selection strategy. Correct data analysis

Reviewer #2: A systematic review is an appropriate study design for synthesizing existing evidence on a specific topic. The objective of this study and the research questions were clearly presented. There are no ethical concerns to mention. However, the major limitation of this review is that, apart from the screening of the titles and abstracts of the English record, which was carried out by two authors, all the other stages of record evaluation, including data collection and article quality assessment, were carried out by a single author. The authors report that only 10% of English records underwent quality control for data collection, without giving any information on the outcome of this control process. The 100% of manuscripts in other languages (apart from English) were evaluated by a single author. "Best practice guidelines for conducting systematic reviews argue that the literature search and sifting process is ideally conducted by two separate reviewers, who must both agree on work to be included." (Siddaway et al., 2019; PMID: 30089228). Given the complexity and diversity/abundance of information reported in this review, this raises serious questions about the quality/validity of the data presented in this manuscript.

Reviewer #3: Though I admit my lack of familiarity with the specific tools used to assess bias in the presented study, on cursory investigation it seems the approach taken by the authors is valid. The authors have not stated a goal for an exhaustive review of the literature, and what new data they were able to identify in their review constitute a reasonable collection of newer methods since the previously described Cardona-Arias et al review they cite. 

They specifically sought to describe the utility of these tests as measured by their accuracy for disease of various forms and locations, as well as the localization of disease. They appear to have devised a reasonable approach to characterizing disease stage and location reporting from the studies included in their review. While the precise patient characteristics across all reviewed studies were challenging to unify due to heterogeneity of included studies, I think the authors did address this adequately in their discussion, and helpfully proposed a framework for future reporting in the literature within this space which might aid future efforts to synthesize recent work.

 The basis of excluding subgroups with a sample size of <20 in a given record (lines 229-230) from analysis for a given test is not entirely clear since narrative analysis was chosen over meta-analysis. Is this related to some concern for bias? Statistical consideration? A short discussion of the associated statistical methods may be of interest.

**Results**

-Does the analysis presented match the analysis plan?

-Are the results clearly and completely presented?

-Are the figures (Tables, Images) of sufficient quality for clarity?

Reviewer #1: Data well presented with sufficient number of figures and tables. Supplementary files provided.

Reviewer #2: The data were described as planned. Meta-analysis was not performed due to the heterogeneity of the studies. 

The tables are large, with a lot of information and abbreviations that make reading fastidious. Figures are of poor quality (low resolution).

Reviewer #3: My gestalt reading through the results was that the intended analysis was successfully performed, though the inconsistent way the results were presented left me needing to read a few times before realizing the complete dataset for sensitivity and specificity was in S2. As an example, in the case of the LLGP WB, the authors report good performance of the test for multiple, active parenchymal cysts, but go on to discuss sensitivity for single cysts, then good specificity of the test (lines 293-296). For B158/B60 and TsW8/TsW5, they report sensitivities (lines 298-299). For the urine antigen tests, they report accuracy (lines 300-301). In their discussion the authors highlight the promising HP10-Ag LFA with 75-100% sensitivity based on group (lines 448-450). I suggest that consistently reporting all three values in the narrative text or choosing one (e.g. accuracy, Sn or Sp) to report with guidance to consult Table 1 or S2 for the others not reported would make the presented results clearer.

Line 366 is at odds with results presented in lines 272-3 (504 evaluated tests reported) and 276-7 (53 studies reported as included comprising 123 tests evaluated)

Line 403 figure reference: did this mean Fig 3 Q4 (concerning RoB analysis)? 

It is less clear why some of the records previously evaluated by Cardona-Arias (e.g. Villota G, 2003; Rao V, 2011; Cho S, 1986; etc.) did not meet inclusion criteria for this study as they otherwise would appear to have met the prior review's requirements for inclusion and meta-analysis. It may satisfy only my own curiosity, but it would be interesting supplementary material (if possible within limits for this article submission) to also see a list of the studies excluded after full-text review.

**Conclusions**

-Are the conclusions supported by the data presented?

-Are the limitations of analysis clearly described?

-Do the authors discuss how these data can be helpful to advance our understanding of the topic under study?

-Is public health relevance addressed?

Reviewer #1: Discussion addressing most important findings. Clear conclusions and recommendations provided

Reviewer #2: The conclusion of the manuscript is supported by the data presented, and the public health relevance of the study was highlighted. The authors innovated by proposing some information to be taken into account to improve the STARD 2015 reporting guideline for diagnostic accuracy studies assessing immunodiagnosis of neurocysticercosis. However, a specific paragraph summarizing the limitations of the study has not been included and could be added.

Reviewer #3: I feel that the authors did convey a good understanding of limitations of the analysis they performed, as well as some of the underpinnings limiting that analysis, which increases my confidence in the validity of their conclusions.

I believe the authors successfully make the case that NCC is of public health importance, that a good understanding of test performance is warranted to improve the prospect of diagnosis in the community, that consistent interpretation of this performance can be challenging because of heterogeneous data reporting in the literature, that there are some promising testing modalities available or in development, and that future study of NCC diagnostics would benefit from a standardized approach to results reporting.

**Editorial and Data Presentation Modifications?**

Reviewer #1: No modifications suggested

Reviewer #2: N/A

Reviewer #3: I want to emphasize that the article is overall readily intelligible, though there are a few minor areas where copy editing would be “cosmetically” appealing or could enhance clarity (e.g. line 52 “fulfil” [sic] is misspelled; line 373 “synthetization” [sic] is perhaps used in lieu of "synthesis"; lines 420-421 word choice “LLGP Western Blot approximates high sensitivity and specificity aspirations” is not entirely clear in its meaning)

Table 1 label “Par Act 1” mismatches with caption label “par act sing” for single active parenchymal lesions

**Summary and General Comments**

Reviewer #1: Excellent paper providing a comprehensive and critical systematic review on serological methods for diagnosis of NCC. High heterogeneity of studies has caused limitations in making conclusions (no meta-analysis).

My only (small) comment is that the authors have not included tests on CSF in their study, while this matrix is regularly used in the diagnosis of NCC. It may be appropriate to add a short note in the discussion (or introduction) section on why this matrix was not included. Collecting CSF is more invasive than blood sampling, and therefore less practical to use in field conditions.

Reviewer #2: See comment file.

Reviewer #3: Neurocysticercosis is a clinically important but neglected disease on a global scale. While gold-standard diagnostics have been described, these may be onerous from the standpoint of cost or expertise required to deploy them consistently in more resource-limited settings, and so alternative and less invasive diagnostics are of value. While other reviews have been performed in the not-too-distant past, these have synthesized available data for each of three broad strategies (EITB, Ag-detection ELISA, and Ab-detection ELISA) and have not evaluated performance of the meta-analyzed studies. This prior review also represented a smaller number of studies all occurring at least a decade ago. Unsurprisingly, as technologies have advanced there have been numerous new modalities described in recent years. Our authors sought to systematically review immunodiagnostics utilizing urine and serum sources and their performance in the detection of disease, where possible specifically analyzing their performance for the detection of disease according to its location and disease stage. 

Though heterogeneity of component data made a side-by-side comparison of studies in the literature challenging, they have done a nice job of collating recent advancements in the immunodiagnostic space and offering at least a basic descriptive narrative of how these tests perform in certain clinical scenarios. Based on their findings, it appears that there are some more promising tests which perform well in the initial diagnosis of disease. Our authors rightly point out a need for a consistently reliable way to track disease progress and the value that would come from a more standardized approach to reporting of diagnostics in future.

PLOS authors have the option to publish the peer review history of their article (what does this mean?). If published, this will include your full peer review and any attached files.

Reviewer #1: No

Reviewer #2: Yes: Fiston Ikwa ndol Mbutiwi

Reviewer #3: No

Figure Files:

Data Requirements:

Reproducibility:

References

---

## [Editor Report · Decision Letter 1]

21 Oct 2024

Dear Ms. Van Acker,

We are pleased to inform you that your manuscript 'Accuracy of immunological tests on serum and urine for diagnosis of *Taenia solium* neurocysticercosis: A systematic review' has been provisionally accepted for publication in PLOS Neglected Tropical Diseases.

Best regards,

Christine M. Budke

Academic Editor

Francesca Tamarozzi

Section Editor

---

## [Editor Report · Acceptance letter]

5 Nov 2024

Dear Ms. Van Acker,

We are delighted to inform you that your manuscript, "Accuracy of immunological tests on serum and urine for diagnosis of *Taenia solium* neurocysticercosis: A systematic review," has been formally accepted for publication in PLOS Neglected Tropical Diseases.

Best regards,

Shaden Kamhawi

co-Editor-in-Chief

Paul Brindley

co-Editor-in-Chief
